# Scale-Invariant, Robust Sparse PCA for Large Data via Differentiable Penalties

## Abstract

Sparse PCA finds low-dimensional structure that loads on few features. Existing methods couple learning and feature selection by applying non-differentiable penalties that force retain-or-zero decisions during optimization. This eliminates features before the optimizer has established which ones matter, limits scalability to serial coordinate-update solvers, and fails when components share support. We introduce DROSS-PCA (Differentiable RObust Scalable Sparse PCA), which decouples learning and feature selection. It learns by optimizing a fully differentiable objective combining robust reconstruction, a smooth sparsity penalty, and an orthogonality term. It selects features post-hoc via cosine-preserving pruning. DROSS-PCA is GPU-accelerated, scaling sparse PCA to large data sets beyond the reach of existing methods. All loss terms are calibrated to be independent of data dimensionality and scale, giving penalty weights consistent meaning across data sets. On synthetic benchmarks, DROSS-PCA matches or outperforms established methods on support recovery. Under outlier contamination it substantially exceeds non-robust methods and remains competitive with dedicated robust methods, which do not scale to large problems. The smooth formulation of DROSS-PCA enables stability analysis via random initializations since no feature is eliminated during training. We find that the method's feature selection is stable when the generating basis is theoretically identifiable and shows increased uncertainty precisely when it is not. On the Human Lung Cell Atlas ($584{,}944 \times 27{,}402$), gene selection is 97% consistent across independent random initializations, and consensus gene modules identified from reproducibility across runs are enriched for known biological pathways. On daily sea-surface temperature fields (ERA5, $23{,}376 \times 484{,}778$), the leading sparse mode reproduces the observed El Niño index and the leading modes localize to individual ocean basins.

## 1 Introduction

Principal component analysis reduces dimensionality by finding directions of maximum variance, but every feature may contribute to every component. In high-dimensional settings such as transcriptomics (Wolf et al., 2018; Stuart et al., 2019) and population genetics (Patterson et al., 2006; Price et al., 2006), this makes components difficult to interpret. A component that loads on thousands of genes identifies a direction of variation without identifying which genes define it. Sparse PCA methods modify the objective so that each component loads on few features, combining dimensionality reduction with feature selection.

Dimensionality reduction and feature selection are two distinct tasks. The sparse PCA methods we survey couple them (Table 1). The optimization objective enforces sparsity so that the solver produces loadings with exact zeros when it terminates. For exact zeros to be stationary points, the penalty must have a kink at the origin, a non-vanishing force toward zero that holds coefficients there once they arrive. This mechanism underlies every penalty designed for exact sparsity, including the lasso (Tibshirani, 1996), SCAD (Fan and Li, 2001), MCP (Zhang, 2010), and hard thresholding (Ma, 2013). Their solvers implement this geometry by applying a binary retain-or-zero decision to each coefficient at each iteration via thresholding or proximal operators (Parikh and Boyd, 2014). Learning and selection are therefore coupled by the mechanics of the solver.

This coupling has two consequences. First, features are selected or discarded before the optimizer has established which ones matter. Correlated features compete for shared variance, and which survive thresholding depends on solver internals (update order, step size) rather than solely on the data (Zou and Hastie, 2005; Freijeiro-González et al., 2022). The problem is compounded when components share support. A feature that contributes to multiple components may be thresholded to zero in some before the optimizer has found a good allocation. Camacho et al. (2020) showed that existing methods fail to recover known sparse structure even on noise-free data when components share features. Second, element-wise coordinate updates and proximal steps are inherently serial, leaving existing implementations CPU-bound and limited to moderate dimensionality (Pedregosa et al., 2011).

We introduce DROSS-PCA (Differentiable RObust Scalable Sparse PCA), which decouples learning and feature selection. DROSS-PCA optimizes a fully smooth objective via minibatch SGD, producing a loading matrix in which signal and noise entries separate by magnitude. Features are then selected post-hoc. No coefficient is set to zero during training, so the optimizer explores the full loading space without premature feature elimination. The smooth objective admits standard GPU-accelerated autodifferentiation.

The objective combines reconstruction, sparsity, and orthogonality terms. All three are normalized so that the sparsity and orthogonality hyperparameters have consistent meaning independent of data scale and dimensionality, allowing the same settings to transfer across data sets. Robust sample weighting is integrated directly into the reconstruction loss, providing outlier resistance without separate preprocessing. Post-hoc feature selection uses cosine-preserving pruning, which controls the angular deviation from each learned direction rather than applying magnitude thresholds.

The smooth formulation enables a stability analysis that existing methods cannot provide. With $\ell_1$ penalties, thresholding makes binary feature decisions at every iteration, so features eliminated early cannot be recovered. This makes random initialization impractical and ties existing methods to PCA initialization. With a smooth penalty, no feature is irrecoverably eliminated during training, so different random initializations can meaningfully explore the solution space. By fitting from many independent initializations, we measure which structure is consistent across solutions and which varies. On synthetic benchmarks where the identifiability of the generating structure is controlled by design, we observe that empirical stability mirrors theoretical identifiability (Lei and Vu, 2015). Feature selection is near-deterministic when the generating basis is identifiable (disjoint support) and degrades precisely when shared support makes the basis non-identifiable. This correspondence, observed but not formally proven, suggests that the variation across initializations reflects properties of the data rather than artifacts of the optimizer.

We validate on synthetic benchmarks using the spiked covariance model (Johnstone and Lu, 2009) in a factorial design that crosses support overlap with eigenvalue separation at three signal-to-noise levels. The method stably identifies active features across all configurations but shows increased uncertainty in component assignments when the generating basis is theoretically non-unique. On the Human Lung Cell Atlas, a large single-cell data set beyond the reach of existing sparse PCA methods, gene selection is 97% consistent across independent random initializations, and consensus gene modules identified from reproducibility across runs are enriched for known biological pathways. On daily ERA5 sea-surface temperature at 0.25° resolution, the leading sparse mode reproduces the observed El Niño–Southern Oscillation index, and the leading modes localize to individual ocean basins rather than spanning them as dense EOFs must.

## 2 Related Work

We organize the sparse PCA literature along five axes on which methods differ, summarized in Table 1: whether feature selection is *deferred* until after learning, whether the method *scales* to large data sets, whether it is *robust* to outliers, whether its penalty weights are *calibrated* to a transferable scale, and whether it supports an *initialization-based stability* diagnostic. We write $n$ for the number of samples, $p$ for features (variables), and $k$ for components (the learned directions).

**Coupled feature selection.** Most sparse PCA methods impose sparsity during optimization, coupling learning and selection. The regression/elastic-net reformulation (Zou et al., 2006), regularized low-rank approximation (Shen and Huang, 2008), penalized matrix decomposition (Witten et al., 2009), SCoTLASS

(Jolliffe et al., 2003), and iterative thresholding (Ma, 2013; Johnstone and Lu, 2009) all enforce exact zeros through an $\ell_1$-type penalty or thresholding step applied at every iteration. Selection therefore happens before the optimizer has resolved which features matter, the failure mode analyzed in Section 1. DROSS-PCA trains a dense loading matrix under a smooth objective and prunes post-hoc, so no feature is zeroed during learning.

**Scalability.** Several families are intrinsically limited to modest dimension. The semidefinite relaxation (d'Aspremont et al., 2007) and the certifiable combinatorial formulations (Berk and Bertsimas, 2019; Bertsimas et al., 2022; Del Pia et al., 2025) operate on the $p \times p$ covariance matrix and target one component at a time. The Fantope relaxation (Vu et al., 2013) and its stochastic variant (Qiu et al., 2023) optimize a $p \times p$ projection, incurring an $O(p^3)$ eigendecomposition per step that is already prohibitive at the $p = 27{,}402$ of our single-cell application. The latter uses minibatches but does not remove this per-step cost. Coordinate-descent implementations such as MiniBatch SparsePCA (Pedregosa et al., 2011) and variable projection (Erichson et al., 2020) process minibatches of samples but remain CPU-bound and are not demonstrated beyond moderate $p$. DROSS-PCA optimizes the $p \times k$ loading matrix directly at $O(bpk)$ per step ($b$ the minibatch size), avoiding the per-step $O(p^3)$ and $O(p^2)$ storage costs above and admitting GPU execution.

**Robustness.** A separate line of work combines sparsity with outlier resistance (Croux et al., 2013; Hubert et al., 2016; Pfeiffer et al., 2025; Puchhammer et al., 2026), differing both in the corruption model assumed and in how robustness is achieved. ROSPCA (Hubert et al., 2016) and sPCAgrid (Croux et al., 2013) target casewise (whole-sample) outliers—the model of our contamination experiments—whereas SCRAMBLE (Pfeiffer et al., 2025) targets cellwise (individual-entry) corruption. The two casewise methods take opposite routes to robustness: ROSPCA runs a robust estimation stage first, identifying outliers and a robust subspace, and only then computes sparse loadings, whereas sPCAgrid builds robustness into the objective itself, maximizing a robust measure of scale under an $\ell_1$ penalty. Neither, however, has been demonstrated at the tens-of-thousands-of-features scale of our single-cell data (Appendix G). Because their casewise model matches our contamination setting, we use ROSPCA and sPCAgrid as the robust baselines in the synthetic evaluation and report the comparison in detail (Section 4.1), but their cost precludes running them at our single-cell scale. DROSS-PCA instead builds robustness directly into its differentiable reconstruction loss, through a per-sample tanh weighting that smoothly down-weights high-residual samples (Section 3.1.2), so outlier resistance is retained at the larger problem sizes these methods do not reach.

**Penalty calibration.** Because $\ell_1$ and related penalties are expressed in the raw units of the data, their optimal weights depend on $p$, $n$, and the data scale and must be retuned per data set. DROSS-PCA normalizes each additive penalty by the Eckart–Young–Mirsky rank-$k$ reconstruction floor (Section 3.1.5), which makes the sparsity and orthogonality weights dimensionless and gives them consistent meaning across data sets of different size and scale (the *Calib.* column of Table 1).

**Initialization stability.** DROSS-PCA supports a component-level stability diagnostic obtained by re-fitting from many random initializations and measuring which structure recurs. By contrast, convex formulations with a unique optimum have nothing to vary across restarts. Stability selection (Meinshausen and Bühlmann, 2010; Sill et al., 2015) also assesses reproducibility, but it resamples the data and records how often each feature is selected—one frequency per feature—rather than tracking whether features recur in the *same* component across runs. It therefore does not probe the stability of the learned components (the *Init. stab.* column).

## 3 Methods

DROSS-PCA is a two-stage pipeline (Figure 1A). *Learning* (Section 3.1) optimizes a fully smooth objective to produce loadings in which signal and noise entries separate by magnitude, and *feature selection* (Section 3.2) imposes exact zeros post-hoc via cosine-preserving pruning. Because the smooth objective supports arbitrary initialization, fitting the pipeline from multiple independent random initializations provides a stability analysis (Figure 1B, Section 3.3) that identifies empirically stable structure in the decomposition. The

Table 1: Capability comparison across sparse PCA method families. ✓ = yes, (✓) = partial, × = no. **Deferred**: trains a dense loading matrix and zeros entries only afterward. **Scales**: GPU/minibatch usable at $p \gtrsim 10^4$. **Robust**: built-in outlier resistance. **Calib.**: penalty weights normalized to a transferable, dimensionless scale. **Init. stab.**: identifiability diagnostic from random initializations at the component level.

| Method / family (representative) | Deferred | Scales | Robust | Calib. | Init. stab. |
|---|---|---|---|---|---|
| SDP relaxation (d'Aspremont et al., 2007) | × | × | × | × | × |
| Certifiable MIP (Berk and Bertsimas, 2019; Bertsimas et al., 2022; Del Pia et al., 2025) | × | × | × | × | × |
| SCoTLASS (Jolliffe et al., 2003) | × | × | × | × | × |
| SPCA / elastic net (Zou et al., 2006) | × | × | × | × | × |
| Regularized SVD (Shen and Huang, 2008) | × | × | × | × | × |
| PMD / SPC (Witten et al., 2009) | × | × | × | × | × |
| MiniBatch SparsePCA (Pedregosa et al., 2011) | × | (✓) | × | × | × |
| Variable projection (Erichson et al., 2020) | × | (✓) | (✓)$^{\dagger}$ | × | × |
| Iterative thresholding (Ma, 2013; Johnstone and Lu, 2009) | × | × | × | × | × |
| Fantope FPS (Vu et al., 2013) | × | × | × | × | × |
| Gradient / online (Qiu et al., 2023) | × | (✓)$^{\ddagger}$ | × | × | × |
| Robust sparse PCA (Croux et al., 2013; Hubert et al., 2016; Pfeiffer et al., 2025) | × | × | ✓ | × | × |
| Stability selection (Meinshausen and Bühlmann, 2010; Sill et al., 2015) | × | × | × | × | (✓)$^{\S}$ |
| **DROSS-PCA (ours)** | ✓ | ✓ | ✓ | ✓ | ✓ |

$^{\dagger}$ a robust variant exists but targets cellwise corruption, a different model from our casewise benchmark, so we use the standard variant. $^{\ddagger}$ minibatch/online, but optimizes a $p \times p$ projection ($O(p^3)$/step), prohibitive at the single-cell dimension $p = 27,402$. $^{\S}$ stability diagnostic, but by resampling the data at the feature level, not from initializations at the component level.

pruning stage makes discrete retain-or-zero decisions, so the pipeline as a whole is not end-to-end differentiable. The differentiable part is the learning objective, whose reconstruction, sparsity, and orthogonality terms are all smooth. We utilize this smoothness in two ways: it admits standard GPU-accelerated optimization, and it keeps every feature recoverable throughout training rather than zeroing any, which makes the initialization-based stability analysis possible. Keeping selection out of the differentiable objective is therefore deliberate.

### 3.1 Learning

#### 3.1.1 Objective

Given a data matrix $\mathbf{X} \in \mathbb{R}^{n \times p}$ (rows are samples, columns are features), we seek $k$ loading vectors $\mathbf{W} \in \mathbb{R}^{p \times k}$ that minimize

$$\mathcal{L}(\mathbf{X}, \mathbf{W}) = \mathcal{L}_{\text{recon}}(\mathbf{X}, \mathbf{W}) + \lambda_s \, \mathcal{L}_{\text{sparse}}(\mathbf{W}) + \lambda_{\text{orth}} \, \mathcal{L}_{\text{orth}}(\mathbf{W}), \tag{1}$$

$\mathcal{L}_{\text{recon}}$ measures reconstruction quality. $\mathcal{L}_{\text{sparse}}$ drives small loading entries toward zero. $\mathcal{L}_{\text{orth}}$ penalizes inter-component correlation, maintaining near-orthonormality of $\mathbf{W}$. The objective is optimized by minibatch stochastic gradient descent over batches of $b$ samples drawn from $\mathbf{X}$ and $\mathbf{W}$ is initialized as either the top-$k$ PCA components or random orthogonal vectors.

#### 3.1.2 Robust Reconstruction Loss

Let $\bar{\mathbf{x}} = \frac{1}{n} \sum_{i=1}^{n} \mathbf{x}_i \in \mathbb{R}^p$ denote the column mean. For a centered observation $\tilde{\mathbf{x}}_i = \mathbf{x}_i - \bar{\mathbf{x}}$, the scores are $\mathbf{z}_i = \mathbf{W}^\top \tilde{\mathbf{x}}_i$ and the residual is $\mathbf{r}_i = \tilde{\mathbf{x}}_i - \mathbf{W}\mathbf{z}_i$. In matrix form, the reconstruction of the centered data is $\tilde{\mathbf{X}}\mathbf{W}\mathbf{W}^\top$, which equals the optimal rank-$k$ projection when $\mathbf{W}$ has orthonormal columns.

The reconstruction loss is the weighted mean squared residual normalized by a robust variance estimate $\hat{\sigma}_{\text{rob}}^2$:

$$\mathcal{L}_{\text{recon}}(\mathbf{X}, \mathbf{W}) = \frac{1}{\hat{\sigma}_{\text{rob}}^2} \cdot \frac{1}{bp} \sum_{i=1}^{b} \sum_{j=1}^{p} \bar{w}_i \, r_{ij}^2, \tag{2}$$

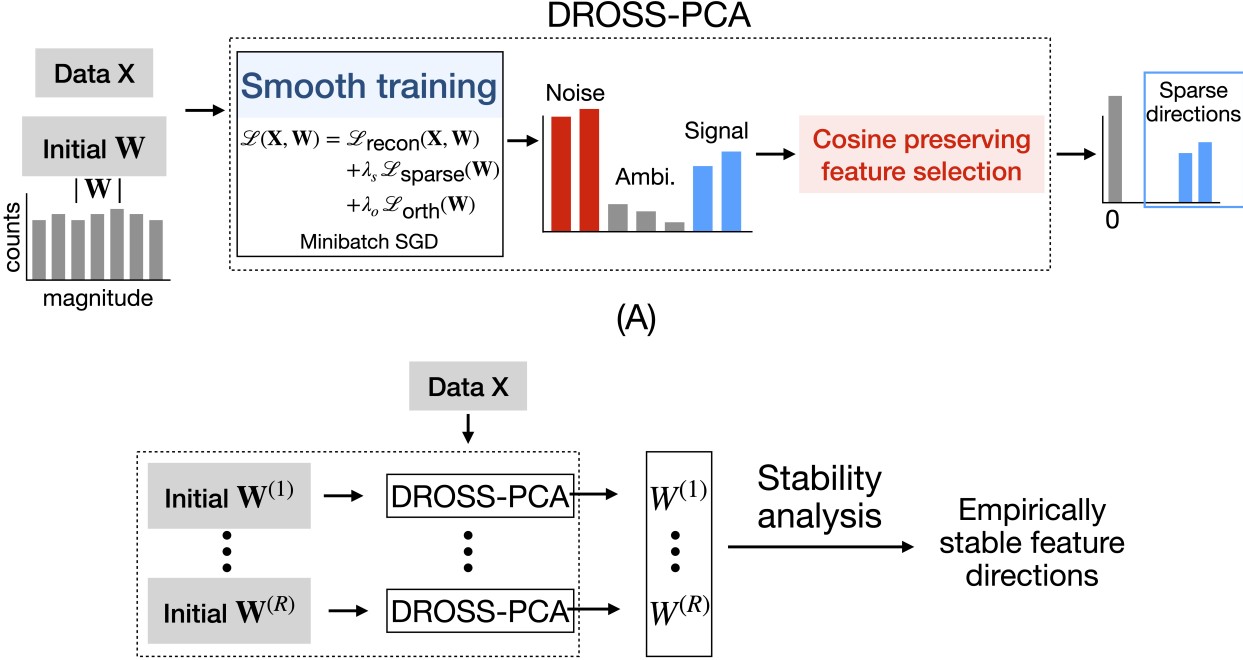

Figure 1: DROSS-PCA overview. **(A)** A single run. Smooth training optimizes a fully differentiable objective via minibatch SGD, producing loadings in which signal, ambiguous, and noise entries separate by magnitude. Cosine-preserving pruning then zeros the smallest entries per component while bounding the angular deviation from the learned direction, yielding sparse directions. No coefficient is set to zero during training. **(B)** Stability analysis. The pipeline is run from $R$ independent random initializations on the same data, producing $R$ pruned loading matrices $\mathbf{W}^{(1)}, \ldots, \mathbf{W}^{(R)}$. Comparing these identifies which features and component assignments are consistent across runs.

where $\bar{w}_i$ downweights outlying samples and the division by $\hat{\sigma}_{\mathrm{rob}}^2$ normalizes the loss to $O(1)$ regardless of data scale. We define $\hat{\sigma}_{\mathrm{rob}} = 1.4826 \cdot \mathrm{median}_{j:\,\mathrm{MAD}_j > 0}(\mathrm{MAD}_j)$, where $\mathrm{MAD}_j = \mathrm{median}_i |x_{ij} - \bar{x}_j|$ (see Appendix A.2 for details).

We define the per-sample robustness weight $\bar{w}_i$ as follows. Within each minibatch, we compute the root-mean-square residual $e_i$, normalized by the robust scale $\hat{\sigma}_{\mathrm{rob}}$, and derive weights via a tanh function:

$$e_i = \sqrt{\frac{1}{p\,\hat{\sigma}_{\mathrm{rob}}^2} \sum_{j=1}^{p} r_{ij}^2}, \qquad w_i = \frac{\tanh(e_i)}{e_i}, \qquad \bar{w}_i = \frac{w_i}{\frac{1}{b}\sum_{i=1}^{b} w_i}. \tag{3}$$

Samples with small residuals receive $w_i \approx 1$ (recovering squared-error behavior), while samples with large residuals receive $w_i \approx 0$ (effectively discarded). Because $e_i$ is measured in units of $\hat{\sigma}_{\mathrm{rob}}$, the outlier downweighting is invariant to the overall scale of the data.

### 3.1.3 Smooth Sparsity Penalty

The sparsity and orthogonality penalties operate on the column-normalized loading matrix $\hat{\mathbf{W}}$, where $\hat{\mathbf{w}}_\ell = \mathbf{w}_\ell / \|\mathbf{w}_\ell\|_2$, decoupling the direction of each component from its magnitude.

The sparsity loss is

$$\mathcal{L}_{\mathrm{sparse}}(\hat{\mathbf{W}}) = \frac{1}{pk} \sum_{j=1}^{p} \sum_{\ell=1}^{k} \phi(\hat{W}_{j\ell};\, \varepsilon), \tag{4}$$

where

$$\phi(x; \varepsilon) = \log\left(1 + \frac{x^2}{\varepsilon^2}\right) \tag{5}$$

is a fully smooth ($C^\infty$) version of the log penalty $\log(|x| + \varepsilon)$ of Candès et al. (2008). The gradient increases from zero at $x = 0$ to a maximum of $1/\varepsilon$ at $|x| = \varepsilon$, then decays as $2/|x|$ for $|x| \gg \varepsilon$. This provides intrinsic adaptive lasso behavior (Zou, 2006) without explicit reweighting. We set $\varepsilon_0 = 1/\sqrt{p}$ since entries of $\hat{\mathbf{W}}$ are $O(1/\sqrt{p})$ at initialization.

### 3.1.4 Orthogonality Penalty

The orthogonality loss is

$$\mathcal{L}_{\text{orth}}(\hat{\mathbf{W}}) = \frac{1}{k^2} \left\| \hat{\mathbf{W}}^\top \hat{\mathbf{W}} - \mathbf{I}_k \right\|_F^2, \tag{6}$$

which penalizes the off-diagonal entries of $\hat{\mathbf{W}}^\top \hat{\mathbf{W}}$ (the cosines of inter-component angles).

### 3.1.5 Optimization

**PCA baseline calibration.** The reconstruction loss of the top-$k$ PCA solution is a lower bound on $\mathcal{L}_{\text{recon}}$. The objective is additive, but its terms are heterogeneous: $\mathcal{L}_{\text{recon}}(\mathbf{X}, \mathbf{W})$ is a property of the data, its magnitude set by how much variance the rank-$k$ subspace leaves unexplained, whereas the sparsity and orthogonality penalties are functions of the column-normalized loadings $\hat{\mathbf{W}}$ alone. The reconstruction term thus lives in the data space and the penalties in the loading space, on unrelated scales. When this floor is large, the values of $\lambda_s$ and $\lambda_{\text{orth}}$ needed to make the sparsity and orthogonality penalties compete with reconstruction depend on the data. Without calibration, finding effective penalty weights would require data-set-specific exploration. We eliminate this by scaling the penalties by the PCA reconstruction loss $\mathcal{L}_{\text{recon}}^{\text{PCA}} = \mathcal{L}_{\text{recon}}(\mathbf{X}, \mathbf{W}_{\text{PCA}})$, evaluated on a small subsample of batches before training. Scaling by this data-space floor puts the loading-space penalties on the scale of the reconstruction loss, so the sparsity and orthogonality terms stay relevant to reconstruction and all terms of the additive objective remain commensurable across data sets. The calibrated objective is

$$\mathcal{L}(\mathbf{X}, \mathbf{W}) = \mathcal{L}_{\text{recon}}(\mathbf{X}, \mathbf{W}) + \lambda_s \, \mathcal{L}_{\text{recon}}^{\text{PCA}} \, \mathcal{L}_{\text{sparse}}(\hat{\mathbf{W}}) + \lambda_{\text{orth}} \, \mathcal{L}_{\text{recon}}^{\text{PCA}} \, \mathcal{L}_{\text{orth}}(\hat{\mathbf{W}}). \tag{7}$$

Computation details are in Appendix A.1.

**Solver and schedule.** The objective (7) is minimized by minibatch stochastic gradient descent using AdamW (Loshchilov and Hutter, 2019). Training is steps-based (not epoch-based), so all hyperparameters are expressed in gradient steps, making them independent of data set size and batch size. The learning rate $\eta$ follows a two-phase schedule, with linear warmup from $\eta_{\text{max}}/100$ to $\eta_{\text{max}}$, then cosine decay to $\eta_{\text{max}}/10$. After the scheduled $T$ steps, training optionally continues at $\eta_{\text{max}}/10$ with patience-based stopping. Each minibatch gradient step requires $O(Bpk)$ operations, linear in all problem dimensions, and is JIT-compiled for GPU execution. The training procedure is summarized in Algorithm 1 (Appendix A.7).

**Sparsity warmup.** Applying full sparsity pressure from initialization can trap the optimizer in poor local minima before the reconstruction term has established a good subspace. The sparsity term is therefore multiplied by a cosine annealing factor $\alpha(t) = \frac{1}{2}(1 - \cos(\pi t/T_{\text{warm}}))$, which ramps from 0 to 1 over $T_{\text{warm}}$ warmup steps. During warmup, $\varepsilon$ is held fixed at $\varepsilon_0 = 1/\sqrt{p}$.

**Adaptive threshold via Otsu's method.** As training progresses and the sparsity penalty drives small entries toward zero, the distribution of $|\hat{W}_{j\ell}|$ develops a concentrated mass near zero separated from a tail of signal entries. After warmup, we adapt $\varepsilon$ to track the boundary between these populations at intervals of $T_\varepsilon$ steps using Otsu's method (Otsu, 1979), which finds the threshold maximizing between-class variance on the histogram of $|\hat{W}_{j\ell}|$ values. The update is floored at the initial value to prevent $\varepsilon$ from collapsing below the natural scale of $\hat{W}$ entries:

$$\varepsilon \leftarrow \max\left(\varepsilon_{\text{Otsu}}, \, 1/\sqrt{p}\right). \tag{8}$$

The floor at $1/\sqrt{p}$ also bounds the peak sparsity gradient at $\sqrt{p}$, preventing the sparsity term from dominating reconstruction.

**Gradient propagation.** Both the sparsity and orthogonality terms operate on $\hat{\mathbf{W}}$ with columns $\hat{\mathbf{w}}_\ell = \mathbf{w}_\ell/\|\mathbf{w}_\ell\|_2$, but they differ in how gradients propagate to $\mathbf{W}$. For the sparsity term, the column norms $\|\mathbf{w}_\ell\|_2$ are treated as constants (stop-gradient), so $\partial\mathcal{L}_{\mathrm{sparse}}/\partial\mathbf{W}$ acts only on the direction of each column and the optimizer cannot inflate $\|\mathbf{w}_\ell\|_2$ to shrink $|\hat{W}_{j\ell}|$ and weaken the penalty. For the orthogonality term, gradients flow through the full normalization $\hat{\mathbf{w}}_\ell = \mathbf{w}_\ell/\|\mathbf{w}_\ell\|_2$, allowing the optimizer to rotate component directions to reduce inter-component correlation. For the reconstruction term, the per-sample weights $\bar{w}_i$ are detached from the computational graph (stop-gradient on $\bar{w}_i$) to prevent the optimizer from minimizing the loss by manipulating the weights rather than improving the reconstruction (Holland and Welsch, 1977).

**Initialization.** Initialization defaults to the top-$k$ right singular vectors of the centered data via randomized SVD (Halko et al., 2011) (PCA initialization). We also provide random initialization. Draw a random $p \times k$ Gaussian matrix and orthonormalize via QR decomposition to obtain unit-norm orthonormal columns, independent of the data scale.

### 3.2 Feature Selection

**Cosine-preserving pruning (CPP).** The learning stage produces near-orthonormal loading vectors whose directions encode the subspace structure (Figure 1A). Pruning strategies based on magnitude thresholding or percentile cutoffs may deviate the directions from the learned ones. We designed CPP to control the angular deviation directly. For each loading vector, we zero out the smallest-magnitude entries while bounding the cosine distance from the learned direction by a threshold $\theta$ (Algorithm 2, Appendix A.8). Sweeping $\theta$ on a uniform grid from $0°$ to $\theta_{\max}$ (default $10°$, 100 points) yields a one-parameter family of solutions from unpruned to maximally sparse without retraining. Evaluating sparsity and variance explained (VE) at each $\theta$ traces a Pareto frontier. To select a single operating point, we identify the knee of this frontier.

**Operating point selection.** We normalize both axes to $[0,1]$ per curve and smoothen the curve via a Savitzky–Golay filter (Savitzky and Golay, 1964). We compute the rate of change of the tangent angle $\vartheta = \arctan(d\,\mathrm{VE}/d\sigma)$ with respect to normalized sparsity $\sigma$:

$$\frac{d\vartheta}{d\sigma} = \frac{d^2\mathrm{VE}/d\sigma^2}{1 + (d\,\mathrm{VE}/d\sigma)^2}. \tag{9}$$

The operating point is the sparsity at which $d\vartheta/d\sigma$ is most negative, the sharpest transition from flat to steep on the frontier. Compared to arc-length curvature (which uses a $3/2$ exponent in the denominator), this criterion is less sensitive to early gentle bends and selects the onset of substantial degradation rather than the first detectable change.

### 3.3 Stability Analysis via Multiple Initializations

The sparse decomposition may not be uniquely identifiable from the data. With $\ell_1$ methods, thresholding eliminates features irrecoverably at each iteration, making random initialization impractical and tying these methods to PCA initialization. The smooth DROSS-PCA objective has no such constraint, so different random initializations can meaningfully explore the solution space. We fit the pipeline $R$ times from independent random orthonormal initializations, each followed by CPP (Figure 1B), producing $R$ pruned loading matrices $\mathbf{W}^{(1)}, \ldots, \mathbf{W}^{(R)}$. Variation across these matrices reflects two sources that we do not attempt to disentangle: the non-convex optimization may converge to different local minima, and CPP's angular threshold may prune differently depending on the learned loadings. Structure that is consistent across solutions is robustly determined by the data. The consensus procedure below is robust to both sources because features at the pruning boundary are inconsistent across runs and are filtered out.

To measure consistency, we match components across runs via optimal one-to-one assignment (Hungarian algorithm on the Jaccard similarity matrix of component supports, where the Jaccard similarity of two

feature sets is the size of their intersection divided by the size of their union). For each component $j$, we define a *stability frequency* $f_{jg}$ as the fraction of runs in which feature $g$ is active in the matched partner of component $j$. The *consensus core* at threshold $\tau$ is the set of features with $f_{jg} \geq \tau$, and the *stable support* is the union of all consensus cores. At the global level, the pairwise Jaccard similarity of active feature sets across all $\binom{R}{2}$ pairs quantifies overall selection consistency. The full matching procedure is detailed in Appendix D.

Consensus cores provide a per-feature, per-component confidence annotation. Core features ($f_{jg} \geq \tau$) are stably co-assigned across independent solutions, while peripheral features ($f_{jg} < \tau$) reflect ambiguity in the decomposition. This extends stability selection (Meinshausen and Bühlmann, 2010) from binary feature inclusion to component-level assignment. Methods that produce a unique solution by construction cannot provide this diagnostic.

### 3.4 Evaluation Metrics

We evaluate three aspects of each method's output. *Reconstruction quality* is measured by variance explained (VE) at a given sparsity level. Two quantities appear as the axes of the recovery figures and we define them here: *sparsity* is the fraction of exactly zero entries in the loading matrix, and *relative reconstruction error* is $\|\tilde{\mathbf{X}} - \tilde{\mathbf{X}}\hat{\mathbf{W}}\hat{\mathbf{W}}^\top\|_F / \|\tilde{\mathbf{X}}\|_F$ on centered data $\tilde{\mathbf{X}}$ (so $\mathrm{VE} = 1 - (\text{relative reconstruction error})^2$). *Feature recovery* is measured by global area under the precision–recall curve (AUPRC), which ranks features by their row norms across all components and tests whether active features are ranked above inactive ones. Because it is invariant to component permutation and rotation within the loading subspace, global AUPRC is the primary recovery metric. *Orthogonality* is the mean absolute off-diagonal of the direction-normalized Gram matrix $\hat{\mathbf{W}}^\top \hat{\mathbf{W}}$, with lower values indicating less inter-component correlation. Additional metrics including per-component recovery and subspace angles are defined in Appendix B.2.

## 4 Results

We evaluate DROSS-PCA on synthetic benchmarks with known generating structure, testing feature recovery, reconstruction quality, stability, robustness, and computational scaling, and on two large real-world data sets: single-cell transcriptomics from the Human Lung Cell Atlas (HLCA) and daily ERA5 sea-surface temperature fields. Experimental details for synthetic benchmarks are in Appendix B, for the HLCA in Appendix D, and for the sea-surface temperature fields in Appendix E.

We use the spiked covariance model (Johnstone and Lu, 2009) ($p = 500$, $k = 5$, $n = 1,000$):

$$X \sim \mathcal{N}(0, \Sigma), \qquad \Sigma = \mathbf{VDV}^\top + \sigma^2 \mathbf{I}_p, \tag{10}$$

where $\mathbf{V} \in \mathbb{R}^{p \times k}$ has sparse, unit-norm columns, $\mathbf{D} = \mathrm{diag}(d_1, \ldots, d_k)$ contains spike eigenvalues, and $\sigma^2$ is isotropic noise variance. We refer to $\mathbf{V}$ as the *generating basis*, the set of features with at least one nonzero entry in $\mathbf{V}$ as the *generating support*, and columns of $\mathbf{V}$ as the *generating components*. Features nonzero in exactly one generating component are *private* and those in two or more are *shared*.

Baselines include Zou–Hastie SparsePCA (Zou et al., 2006), MiniBatch SparsePCA (Pedregosa et al., 2011), VarPro SPCA (Erichson et al., 2020), and PMD/SPC (Witten et al., 2009), together with the robust sparse PCA methods ROSPCA (Hubert et al., 2016) and sPCAgrid (Croux et al., 2013), each sweeping their sparsity parameter over a grid. All non-robust methods use PCA initialization. The robust baselines use their own robust initialization. For baselines, each hyperparameter configuration produces a model with native sparsity. For DROSS-PCA, results reflect the full pipeline, training followed by CPP operating point selection.

### 4.1 Factorial Design (S1–S4)

A $2 \times 2$ factorial design crosses support overlap (0% vs. 25%) with eigenvalue separation (well-separated vs. degenerate), yielding four structural configurations: S1 (disjoint support, well-separated eigenvalues), S2 (disjoint, degenerate), S3 (overlapping, well-separated), and S4 (overlapping, degenerate). This design

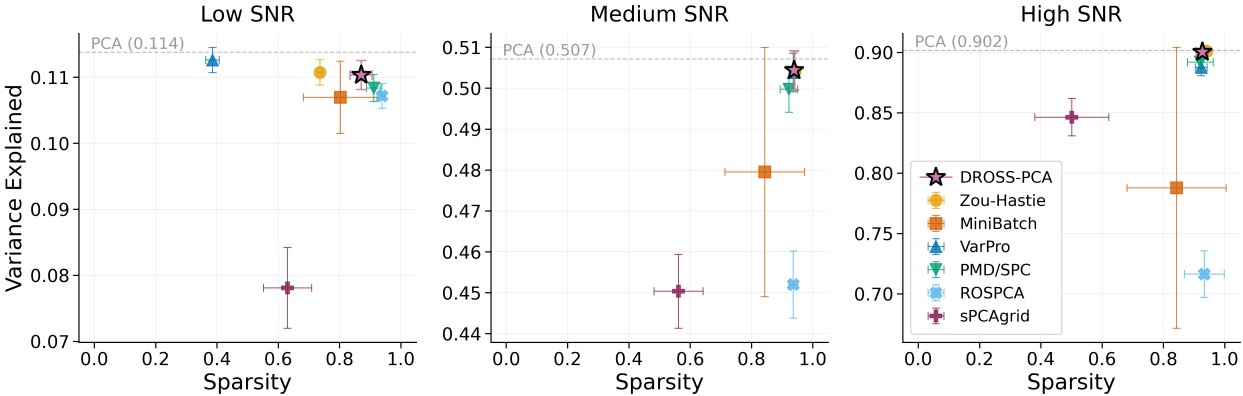

Figure 2: Sparsity vs. variance explained for S1 (disjoint, separated eigenvalues) at three SNR levels. PCA dense ceiling shown as dashed line. DROSS-PCA (⋆) matches the PCA ceiling at above 90% sparsity across all SNR levels. Each point represents one hyperparameter configuration. Error bars show ±1 std over 100 replicates.

is motivated by the identifiability conditions of Lei and Vu (2015), who showed that support overlap and eigenvalue separation jointly determine whether the sparse generating basis is recoverable. Each configuration is crossed with three SNR levels (∼10%, ∼50%, ∼90% variance explained) and 100 replicates, for 1,200 data sets. See Appendix Table 5 for full specifications.

Across all four configurations at high SNR, DROSS-PCA matches the PCA variance-explained ceiling (within ∼0.2%) at above 90% sparsity, with global AUPRC ∼0.96–0.98. The sparse baselines Zou–Hastie (AUPRC ∼0.98), PMD/SPC (∼0.93), and VarPro (∼0.90–0.93) remain competitive on support recovery and sit within a couple percent of the VE ceiling. MiniBatch SparsePCA underperforms on all metrics. The robust baselines pay for their robustness even on this uncontaminated data, but in different ways: ROSPCA's down-weighting leaves its variance explained ∼20–30% below the ceiling, while sPCAgrid instead stays much denser (only ∼50% sparse). Their support recovery nonetheless stays competitive. Results at medium and low SNR follow the same ranking with smaller effect sizes (Appendix Figures 10–16). Figure 2 shows the VE–sparsity frontier for S1, the identifiable baseline (Lei and Vu, 2015).

**S1c: robustness to outlier contamination.** To isolate the contribution of robust sample weighting, we contaminate S1 with a 2 × 2 factorial crossing contamination fraction {2%, 5%} with outlier magnitude {4×, 8×} the data's standard deviation (Appendix Table 6). Since the only difference from S1 is contamination, any performance gap reflects how each method handles corrupted samples.

At mild contamination (2% at 4×), DROSS-PCA and the non-robust baselines recover well at medium and high SNR (global AUPRC ∼0.92–0.98), so robust weighting is unnecessary here. ROSPCA and sPCAgrid trail here (as low as ∼0.83), a consequence of the robustness trade-off they carry on any data rather than of this mild contamination. As contamination increases, the non-robust baselines degrade while DROSS-PCA maintains recovery. At 5% at 8× and low SNR, DROSS-PCA achieves global AUPRC 0.74 while all non-robust baselines remain near ∼0.32, more than a 2× improvement. Figure 3 shows the contamination grid.

**Recovery capability versus operating-point selection.** Under the most severe contamination (8× magnitude, low SNR), the robust baselines behave unlike the non-robust ones: ROSPCA reaches global AUPRC 0.91 at its selected operating point, against DROSS-PCA's 0.55–0.74. This realized gap to ROSPCA reflects operating-point selection rather than recovery capability (Figure 4, for the two 8×/low-SNR cases). DROSS-PCA's recovery is non-monotone in sparsity, peaking near the true sparsity (middle column, dotted line) at 0.88–0.89, comparable to ROSPCA's best of 0.95. However, the reconstruction-based selector does not choose that configuration, because under heavy contamination reconstruction error and support recovery are anti-correlated — fitting the outliers lowers reconstruction error while degrading recovery.

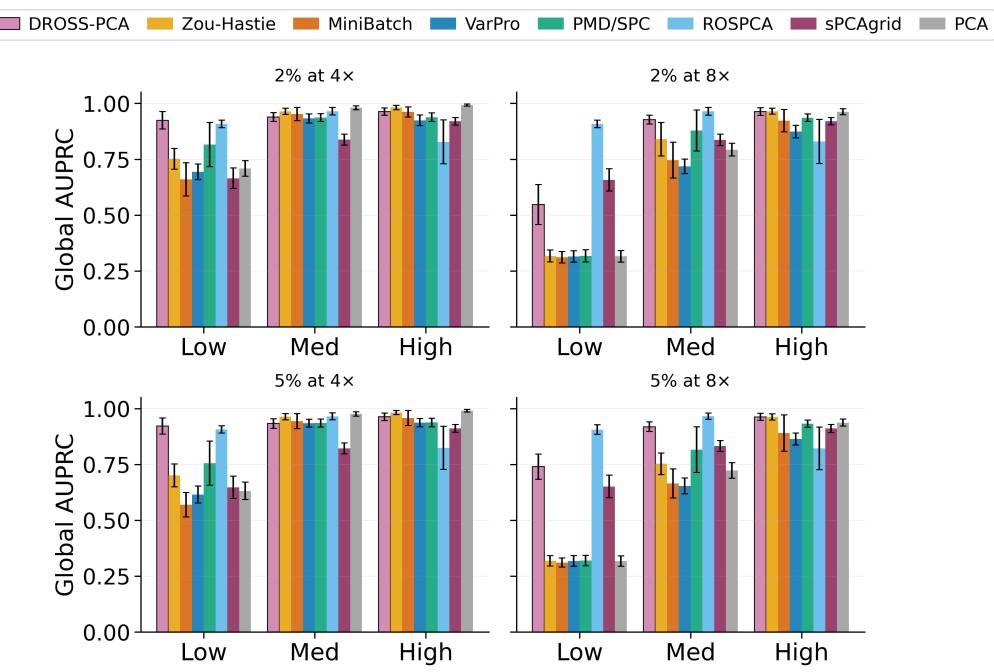

Figure 3: Feature recovery under outlier contamination (S1c). Rows show contamination fraction (2%, 5%) and columns show outlier magnitude (4×, 8×). Each panel shows global AUPRC at three SNR levels. At severe contamination (bottom-right, 8× magnitude, low SNR) the non-robust baselines collapse to global AUPRC ∼0.32 while DROSS-PCA (0.55–0.74) and the robust methods ROSPCA (0.91) and sPCAgrid (∼0.66) retain recovery. The DROSS-PCA–ROSPCA comparison is examined in Figure 4.

ROSPCA does not face this because all of its configurations already lie in the robust, high-reconstruction-error regime, so any selection lands well. Compared at equal reconstruction error, which is comparable across methods (right column), DROSS-PCA's recovery tracks just below ROSPCA's and well above the non-robust methods, which never reach that regime.

**S2: disjoint support and degenerate eigenvalues.** S2 retains the disjoint support of S1 but makes all spike eigenvalues equal. Any rotation within the $k$-dimensional eigenspace is equally optimal for reconstruction, so individual generating components are not identifiable. The appropriate metric is subspace angles, which measure recovery of the column space of $\mathbf{V}$ independently of basis choice. On this metric, DROSS-PCA and Zou–Hastie are equivalent (Figure 12). Apparent differences in component-level metrics reflect basis non-identifiability rather than recovery quality (Appendix B.2).

**S3/S4: overlapping support.** S3 and S4 are nearly indistinguishable on all metrics, indicating that eigenvalue degeneracy adds little cost on top of overlap. Full per-configuration figures are in Appendix (Figures 13–16). Under overlapping support, the generating basis is not uniquely determined by the data (Vu and Lei, 2013; Camacho et al., 2020). We explore this ambiguity empirically via stability analysis in Section 4.3.

## 4.2 Invariance to data scale

Because the sparsity and orthogonality hyperparameters are calibrated to a dimensionless scale (Section 3.1.5), a global rescaling of the data leaves the learned model unchanged. Every term of the objective is scale-free by construction. The reconstruction loss is divided by the robust variance $\hat{\sigma}^2_{\mathrm{rob}}$, and its robust sample weights depend on residuals measured relative to $\hat{\sigma}_{\mathrm{rob}}$ (Equation 3). The sparsity and orthogonality penalties act on the column-normalized loadings. The random initialization is orthonormal and independent of the data scale. Under a global rescaling $\mathbf{X} \mapsto c\,\mathbf{X}$ these factors cancel, so the continuous minimizer is

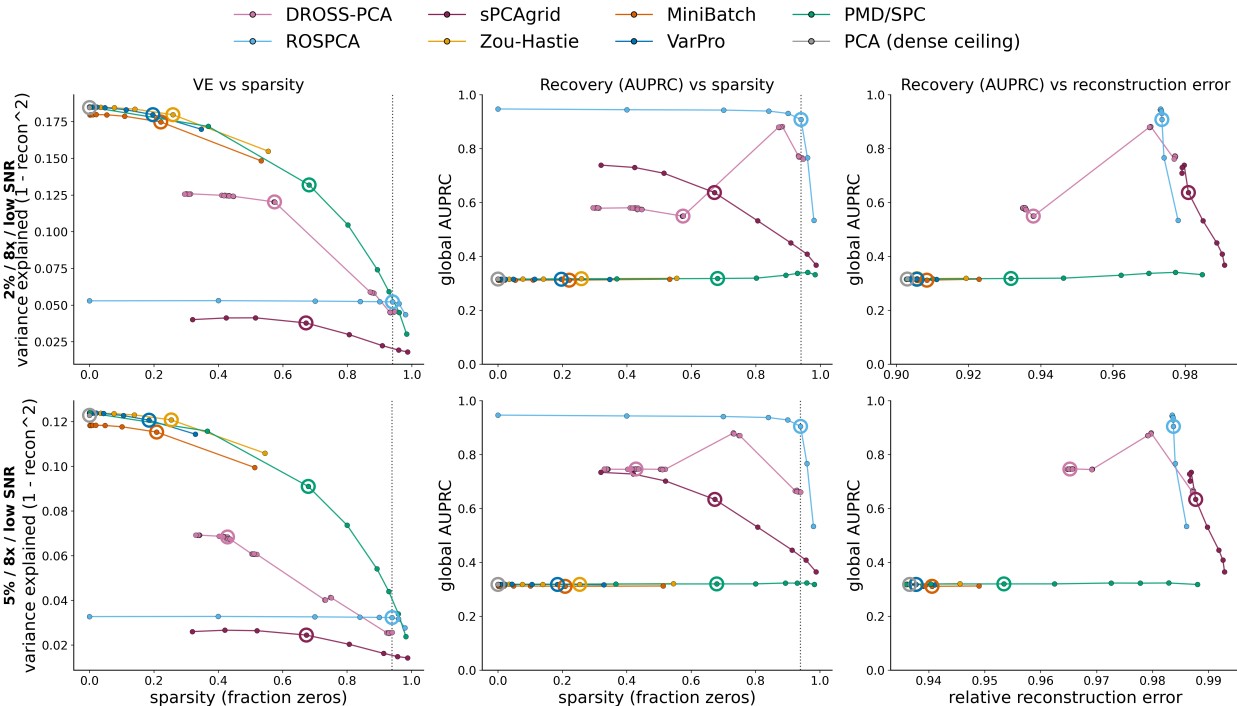

Figure 4: Operating-point selection under severe contamination (S1c, 8× magnitude, low SNR). Rows: top 2%, bottom 5% contamination. Each point is one method configuration averaged over replicates. The open ring marks the automatically selected configuration. Columns: variance explained vs. sparsity, support recovery (global AUPRC) vs. sparsity, and support recovery vs. reconstruction error. Dotted line: true sparsity (470/500). Reconstruction error is comparable across methods, so the right column places all methods on a common axis: DROSS-PCA's recovery peak lies at the same reconstruction level as the robust baselines and above its own selected point, while the non-robust methods never leave the low-reconstruction-error, low-recovery regime.

unchanged for any $c$. Feature selection is a separate, discrete pruning step, whose scale behavior we examine below.

We verify this directly. Holding $(\lambda_s, \lambda_{\text{orth}})$ fixed, we rescale the input by a global factor $c$ spanning six orders of magnitude ($10^{-3}$ to $10^3$) and refit, on a clean data set (S1) and a heavily contaminated one (S1c, where the robust weighting is active). On the clean data all reported quantities—support recovery (global AUPRC), variance explained, selected sparsity, and the operating point—are identical across $c$ to numerical precision. On the contaminated data the fit itself remains nearly invariant—support recovery and variance explained agree to within $4 \times 10^{-3}$ and the learned loadings to under $0.1°$. Because the knee rule selects where the sparsity–variance frontier bends most sharply, on the flat frontier produced by heavy contamination that selection is ill-conditioned and amplifies these tiny fit differences into a shifted pruning threshold and sparsity—always at essentially the same recovery and variance explained. This is the operating-point-selection sensitivity discussed in Section 5. The fit-level invariance holds across the $(\lambda_s, \lambda_{\text{orth}})$ values tested. The penalty weights therefore carry the same meaning regardless of data scale and transfer across data sets without per-data-set retuning. This invariance is to an overall rescaling of the data. The method deliberately does not standardize each feature separately (so higher-variance features contribute more), and is not invariant to arbitrary per-feature rescaling.

### 4.3 Stability across initializations

We apply the multi-initialization procedure (Section 3.3) with $R = 100$ independent random orthonormal initializations per replicate across all four factorial configurations at three SNR levels (Appendix Table 7).

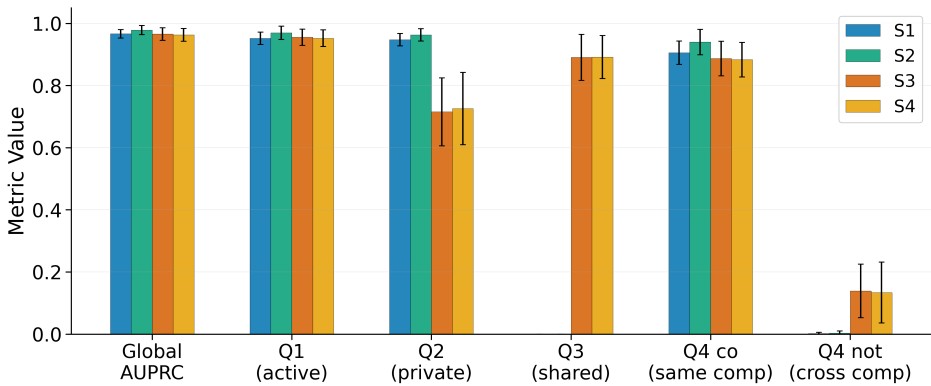

Figure 5: Feature stability at high SNR across the factorial design (mean ± std over 100 replicates, 100 random initializations each). Global AUPRC exceeds 0.96 for all configurations, confirming stable generating support recovery. Q2 (private features placed in exactly one component) drops sharply under overlap (S3, S4) while Q3 (shared features detected as multi-component) is high, indicating that the method trades private-feature precision for shared-feature detection. Q4 co (within-component co-assignment) remains high throughout. Q4 not (cross-component false co-assignment) rises under overlap. S3 and S4 are nearly indistinguishable, indicating that eigenvalue degeneracy adds little ambiguity beyond overlap itself. S1/S2 have no shared features, so Q3 bars are absent. Full SNR breakdown in Appendix Table 8.

S1 and S2 serve as theoretically identifiable controls, while S3 and S4 test the effect of overlap. Figure 5 reports all metrics at high SNR.

**Generating support stability.** Global AUPRC across the 100 solutions exceeds 0.96 for all configurations at high SNR, showing that the generating support is stably recovered regardless of overlap or eigenvalue structure. This is expected since the generating support is the row support of the top-$k$ eigenspace, which is invariant to basis choice.

**Feature assignment stability.** The probability that a private feature is placed in exactly one estimated component (Q2) is $\sim$0.95 under S1 and S2, confirming that eigenvalue degeneracy alone does not degrade feature-level assignments. Under overlap, Q2 drops to $\sim$0.71, a $\sim$24 percentage point decrease reflecting the non-uniqueness of the generating basis. Shared features are reliably detected as multi-component (Q3 $\approx$ 0.89 under S3 and S4), partly explaining the Q2 drop as shared-feature activity spreads into neighboring private features.

**Component-level co-assignment.** Q4 measures whether features are consistently placed in the same component across runs. Within-component co-assignment (Q4 co) remains above 0.88 across all configurations, indicating that features belonging to the same component are stably grouped together. Cross-component false co-assignment (Q4 not) is negligible under S1 and S2 ($<$ 0.01) but rises to $\sim$0.14 under overlap, meaning features from different generating components are sometimes merged into a single estimated component. This is the component-level signature of the ambiguity that overlap introduces. On the HLCA, where no generating basis is available, we apply the full consensus core procedure (Section 3.3) to identify stably co-assigned feature sets within each component (Section 4.5). Table 8 shows the full SNR $\times$ configuration breakdown.

## 4.4 Computational scaling

Figure 6 shows wall-clock time on scaling benchmarks where $n$, $p$, and $k$ increase jointly (Appendix Table 9). Times are averaged across all hyperparameter configurations per method, and DROSS-PCA time includes both fit and CPP. All reported timings use double-precision (float64) arithmetic on both CPU and GPU, so the GPU measurements are directly comparable to the NumPy-based CPU baselines and are not accelerated

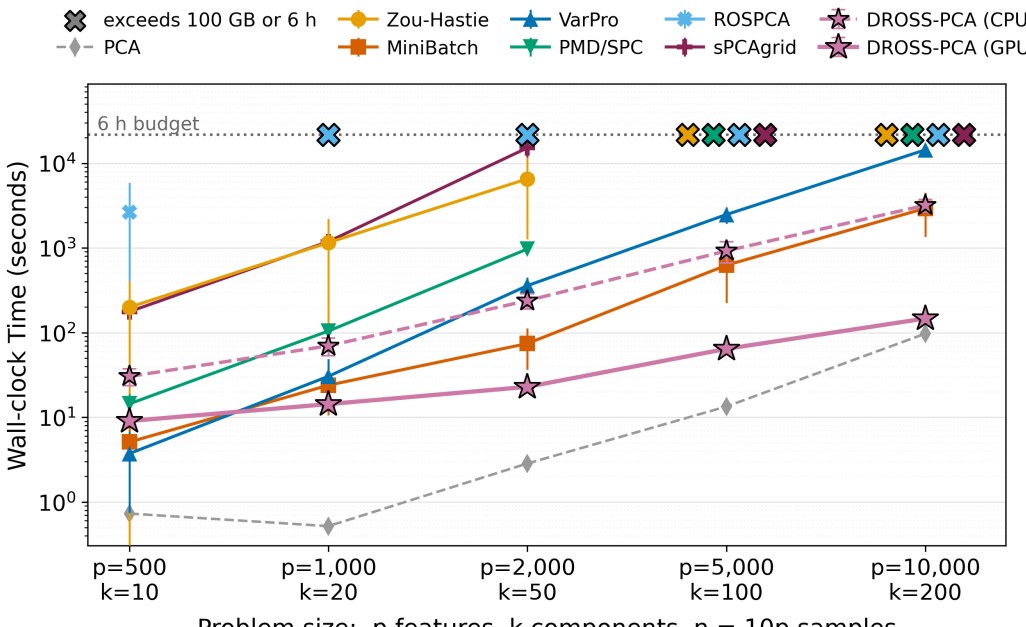

Figure 6: Computational scaling. Wall-clock time (log $y$-axis) versus problem size, dimensions $(p, n, k)$ increasing jointly with $s = 30$ fixed (Table 9). DROSS-PCA is shown on GPU and CPU, with time including fit and CPP. A curve is drawn only through sizes at which *all* of a method's hyperparameter configurations complete (mean $\pm 1$ std). An $\times$ at the budget line marks any size at which at least one configuration exceeds the 100 GB or 6 h budget. DROSS-PCA (GPU and CPU), MiniBatch, and VarPro complete every configuration at every size. The other baselines drop out as $p$ grows: the robust methods (ROSPCA, sPCAgrid) fail earliest, losing configurations by $p = 2000$, while PMD/SPC and Zou–Hastie drop out at the larger sizes ($p \geq 5000$).

by reduced-precision computation. On GPU, DROSS-PCA scales from $\sim 9$ s at $(n, p, k) = (5000, 500, 10)$ to 2.4 min at the largest problem $(100000, 10000, 200)$. On CPU it ranges from 30 s to 53 min and completes every tested size within a 6-hour budget, so the method is usable without a GPU. On GPU the wall-clock grows sublinearly over this range ($16\times$ for a $20\times$ increase in $p$): the minibatch implementation processes fixed-size batches, so per-step cost is dominated by roughly-constant overhead rather than by the full problem size, and this overhead is amortized as the problem grows. The robust baselines do not scale: ROSPCA and sPCAgrid exceed the memory (100 GB) or wall-clock (6 h) budget at every configuration for $p \geq 5000$, and already fail most configurations by $p = 2000$. Their $O(p^3)$ cost places them far below the dimensionality of our single-cell application ($p = 27{,}402$). Among the non-robust baselines, PMD/SPC times out at $p \geq 5000$ and Zou–Hastie drops out by $p = 10000$. MiniBatch and VarPro complete all sizes but require 49 min and 4.0 h at $p = 10000$, versus 2.4 min for DROSS-PCA on GPU. On CPU, DROSS-PCA is slower than MiniBatch in absolute terms at these sizes, but its runtime grows more slowly with $p$ (a shallower slope in Figure 6, bringing the two within 10% at $p = 10000$), and MiniBatch's lower cost comes with a worse sparsity–variance frontier, weaker support recovery, and lower run-to-run consistency (Sections 4.1 and 4.3).

### 4.5 Application: Human Lung Cell Atlas

We apply DROSS-PCA to the Human Lung Cell Atlas (HLCA) core (Sikkema et al., 2023): 584,944 cells $\times$ 27,402 genes, log-normalized single-cell RNA sequencing data. At $k = 5{,}000$ with 100 independent random initializations, gene selection is 97% consistent, consensus gene modules identified from reproducibility across runs are enriched for known biological pathways, and the stable gene support (47% of active genes) carries 98.7% of the model's variance.

This data set is inaccessible to all existing sparse PCA methods because the dense matrix requires $\sim 64\,\text{GB}$ in float32, and all baselines operate on the dense matrix. DROSS-PCA works directly on the sparse representation ($\sim 3\,\text{GB}$), densifying only one minibatch at a time. Full configuration details are in Appendix D.

**Gene selection is stable.**   Figure 7A shows gene selection frequency across 100 random initializations. Of the 27,402 genes, 53% are active in every run, 39% are never active, and fewer than 8% are ambiguous. The mean pairwise Jaccard similarity of active gene sets is 0.97, confirming that gene selection is driven by data structure rather than initialization. The ambiguous genes reflect two sources of variation that we do not attempt to disentangle: different initializations may converge to different local optima, and CPP's angular threshold may prune differently depending on the learned loadings. Rather than isolating these sources, the consensus core analysis below identifies features that are stable regardless of both.

**Component-level stability: consensus cores.**   Gene-level consistency does not address whether genes are assigned to the *same component* across runs. We apply the consensus core procedure (Section 3.3), matching components across runs via optimal one-to-one assignment on Jaccard similarity (Appendix D).

All 5,000 components have non-empty consensus cores at the 90% threshold (Figure 22). Core size declines gradually from 80% to 90% stability and drops sharply above 90%, indicating a natural boundary between stably co-assigned and peripherally assigned genes (Appendix Figure 23).

**Stable cores carry nearly all variance.**   Restricting loadings to the union of all 90%-consensus-core genes (47% of active genes) and recomputing reconstruction yields 98.7% of the full model's variance explained (Figure 7B). The remaining genes collectively contribute less than 1 percentage point. At the 95% threshold, support shrinks further and VE drops to 47%, indicating a sharp transition between stable and ambiguous signal (Appendix Table 11).

**Stable cores are biologically coherent.**   To validate that the reproducible structure corresponds to real biology, we perform over-representation analysis (Fisher's exact test via GSEApy, Fang et al. 2022) on the 90%-consensus cores of the 50 largest components, testing against GO Biological Process, KEGG, and Reactome. All 50 cores show significant enrichment (adjusted $p$-value $< 0.05$), with top-term $p$-values below $10^{-10}$. Notably, the enriched pathways are not generic housekeeping programs but reflect biology specific to lung tissue (Table 2), including cilium assembly (the motile cilia of airway epithelial cells), *Staphylococcus aureus* infection (a leading cause of bacterial pneumonia), antimicrobial humoral response and antigen processing (mucosal immune defense in the respiratory tract), focal adhesion (structural integrity of lung parenchyma), and cytokine response pathways characteristic of the pulmonary immune environment. Gene modules identified from reproducibility across runs, without any biological input, recover lung-specific pathways, providing independent validation that the consensus cores capture tissue-relevant structure rather than generic transcriptional noise. To confirm this enrichment reflects the recovered modules rather than gene-set size, we compared each core against three size-matched baselines under the identical test (Appendix D, Table 10): random gene sets, PCA top-loading genes, and highly variable genes (Wolf et al., 2018). Random sets of matched size yield essentially no enrichment (median 0 significant terms vs. 487 for the cores). Every core exceeds the 95th percentile of its random null (empirical $p$-value $< 0.005$). The cores are comparably enriched to dense PCA top-loadings (median 487 vs. 522) and more enriched than highly variable genes (487 vs. 386), while—unlike a truncation of dense PCA loadings or a single global variability ranking—remaining sparse, component-resolved, and reproducible across initializations.

### 4.6   Application: Sea-Surface Temperature Fields

We apply DROSS-PCA to daily sea-surface temperature (SST) from the ERA5 reanalysis (Hersbach et al., 2020), distributed through WeatherBench-2 (Rasp et al., 2024). This is a second large real-data application, in a domain unrelated to the single-cell setting of Section 4.5, and one in which the leading mode of variability has an accepted physical reference—the El Niño–Southern Oscillation (ENSO)—so recovery can be checked against a quantity defined independently of any PCA. We analyze the full 23,376-day record (1959–2022) at 0.25° resolution ($1440 \times 721$, longitude $\times$ latitude), giving a feature dimension of $p = 484{,}778$ ocean grid cells after masking. Fields are deseasonalized against the shipped daily climatology, restricted to a never-ice

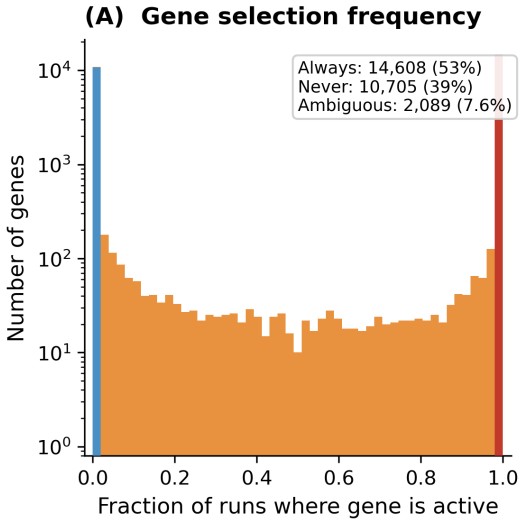
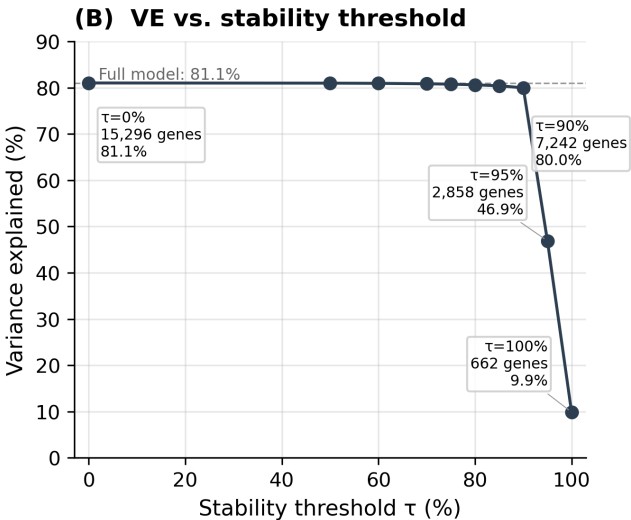

Figure 7: HLCA stability across 100 random initializations at $k = 5,000$. (A) Gene selection frequency. The distribution is strongly bimodal: 53% of genes are active in every run (red), 39% are never active (blue), and only 8% are ambiguous (orange). (B) Variance explained vs. stability threshold $\tau$. The stable support (support-90, 7,242 genes, 47% of active genes) retains 98.7% of the full model's VE, with a sharp cliff between $\tau = 90\%$ and $\tau = 95\%$. Core size distribution in Appendix Figure 22.

Table 2: Functional enrichment of selected consensus cores (core-90). Over-representation analysis (Fisher's exact test) against GO Biological Process, KEGG, and Reactome (background: 14,990 active genes). All 50 top components are significantly enriched. Selected examples span distinct lung-relevant biological programs.

| Core size | Top GO BP | Top KEGG / Reactome | Adj. $p$ |
|---:|---|---|---:|
| 1,869 | Cytoplasmic Translation | Ribosome (KEGG) | $8.3 \times 10^{-40}$ |
| 1,727 | Cilium Assembly | Cilium Assembly (Reactome) | $9.7 \times 10^{-35}$ |
| 634 | Antimicrobial Humoral Response | Immune System (Reactome) | $4.1 \times 10^{-25}$ |
| 551 | Defense Response to Bacterium | Antigen processing (KEGG) | $4.3 \times 10^{-11}$ |
| 492 | Granulocyte Chemotaxis | *S. aureus* infection (KEGG) | $2.5 \times 10^{-12}$ |
| 410 | Regulation of Cell Migration | Focal adhesion (KEGG) | $3.2 \times 10^{-14}$ |
| 376 | Cellular Response to Cytokine Stimulus | Antigen processing (KEGG) | $7.4 \times 10^{-12}$ |

ocean mask, linearly detrended per cell, and area-weighted. The working matrix is the covariance form used throughout (Appendix E).

At the finest resolution the dense $p \times p$ covariance is never formed—at $p = 484{,}778$ it would occupy ~1.9 TB—so the dense-EOF reference and the DROSS-PCA initialization are both obtained by a matrix-free randomized SVD, and the fit densifies only one minibatch at a time, holding the full 45 GB matrix once in host memory (Appendix E). Unlike the sparse single-cell count matrix of Section 4.5, this data matrix is fully dense, so this application also shows that DROSS-PCA's scalability does not rely on input sparsity. We fit $k = 20$ components with robust weighting in single precision.

**The leading mode recovers ENSO.** We match components to the observed Niño-3.4 index by pattern, taking the component whose principal-component time series has the largest absolute correlation $|r|$ with the index (sparse components are not variance-ordered, so the ENSO mode need not come first). This $|r|$ measures how closely the recovered temporal signal follows the observed index, with $|r| = 1$ a perfect match. DROSS-PCA's best-matching component reaches $|r| = 0.977$, against 0.923 for the leading dense EOF, so the recovered mode is essentially the observed ENSO signal. Its loading reproduces the canonical spatial signature of ENSO (Rasmusson and Carpenter, 1982; Deser et al., 2010): the equatorial-Pacific *cold tongue*, the narrow equatorial band where ENSO sea-surface-temperature anomalies are strongest and a

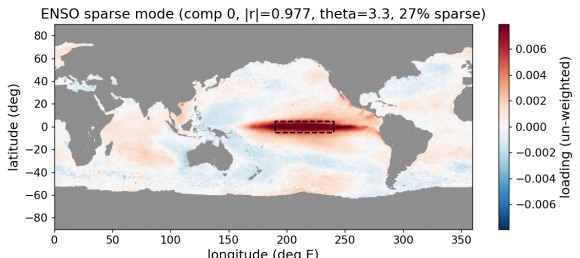 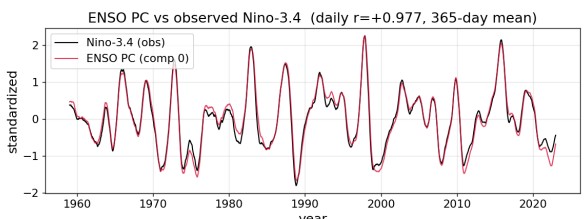

Figure 8: ENSO recovery from daily ERA5 SST at $0.25°$ ($1440 \times 721$). (A) The DROSS-PCA loading of the ENSO-matching component (pruned, area-unweighted): a sparse equatorial-Pacific cold-tongue pattern. The dashed box is the Niño-3.4 region. (B) The component's principal-component time series against the observed Niño-3.4 index (both standardized, 365-day running mean). The daily correlation is $|r| = 0.977$.

long-established feature of the phenomenon, confined here to the Niño-3.4 region (Figure 8). Because the index is defined independently of any PCA, this is a direct check that the recovered mode is the physical ENSO signal rather than an artifact of the decomposition.

**Sparse modes localize to individual basins.** Dense EOFs are forced by mutual orthogonality to spread weight across disconnected ocean basins. The DROSS-PCA loadings are not, and the leading modes instead resolve into individually localized patterns—an equatorial-Pacific ENSO tongue, a North-Pacific (PDO-like) pattern, and an Atlantic center among them (Figure 9). We quantify localization by the effective footprint of each loading (the inverse-participation-ratio effective cell count converted to $km^2$, defined in Appendix E). The sparse modes are $2.82\times$ more localized than the dense EOFs (mean footprint $2.5 \times 10^7$ vs. $7.1 \times 10^7\,km^2$). This is the de-mixing that a deferred, uncoupled sparsity penalty is meant to provide: each component names a region rather than a global contrast.

**Sparsity costs almost no variance.** Cosine-preserving pruning (CPP) traces the sparsity–variance frontier of the learned loadings against the dense-EOF ceiling (Figure 24, Appendix). The selected operating point removes 27% of the loadings at a variance explained of 47.7%, against a $k = 20$ dense-EOF ceiling of 49.2%—a fraction of a percentage point of variance for a substantial reduction in support. The sub-50% figure is not a limitation of sparsity: the dense-EOF subspace at the same $k$ reaches only 49.2%, because a daily $0.25°$ grid resolves far more high-rank weather variance (raising $k$ to 50 recovers 66.1%). The leading large-scale climate modes remain fully within the $k = 20$ subspace.

## 5 Discussion

DROSS-PCA demonstrates that sparse PCA does not require sparsity during training. A smooth objective produces loadings in which signal and noise entries separate by magnitude, and post-hoc pruning imposes zeros at any desired sparsity level from a single trained model. This decoupling enables GPU-accelerated minibatch optimization, scaling sparse PCA to data set sizes ($584{,}944 \times 27{,}402$) that existing methods cannot reach.

**Addressing known failure modes.** Camacho et al. (2020; 2021; 2025) showed that $\ell_1$ methods fail when components share features, because thresholding cannot represent graded contributions to multiple components and deflation propagates errors across components. DROSS-PCA avoids both failure modes. The smooth penalty retains graded loadings throughout training, and all $k$ components are optimized simultaneously. CPP is applied independently per component after convergence, so pruning one component cannot corrupt another.

**Stability as a diagnostic.** The multi-initialization analysis reframes the stability question from "does the method converge to the same solution?" to "what does the data determine?" On synthetics, feature

Leading sparse modes (one basin each) -- 1440x721, k=20

Figure 9: De-mixing at 0.25°. The leading DROSS-PCA modes (pruned, area-unweighted, ordered by variance explained) each localize to a single ocean basin, whereas dense EOFs spread across basins to satisfy orthogonality. Titles give the auto-labelled basin, the effective footprint ($km^2$), and the absolute Niño-3.4 correlation.

assignments are 95% consistent when the generating structure is identifiable, and the ~24 percentage-point cost of overlapping support is attributable to the support structure itself, not the method. On the HLCA, consensus gene modules identified from reproducibility across runs are enriched for lung-specific biological pathways, providing independent validation that the reproducible structure is biologically meaningful.

**Relationship to closest prior work.** The Fantope relaxation (Vu et al., 2013) and its stochastic variant (Qiu et al., 2023) are methodologically closest, as both use minibatch gradient descent for sparse PCA targeting genomics. Three differences are substantive. First, the Fantope framework retains $\ell_1$ constraints, so binary feature selection occurs during optimization. Second, it optimizes a $p \times p$ projection matrix ($O(p^3)$ per step for eigendecomposition), while DROSS-PCA optimizes the $p \times k$ loading matrix directly ($O(bpk)$ per step). Third, the Fantope's convexity precludes using multiple initializations to probe identifiability. Section 2 situates DROSS-PCA against the broader sparse PCA literature. Further detail on robust and combinatorial methods is in Appendix G.

**Limitations and future work.** DROSS-PCA requires GPU for competitive wall-clock times, though a CPU fallback remains faster than most baselines at large $p$. The objective is non-convex, so there are no global optimality guarantees, though the multi-initialization analysis provides empirical evidence that solution quality is data-determined. $\lambda_s$ and $\lambda_{orth}$ are dimension-independent (Section 3.1.5), but optimal values still depend on the desired sparsity level and signal structure. Cosine-preserving pruning yields a well-defined sparsity–variance frontier that bounds each loading's angular deviation from its learned direction (up to $\theta_{max}$), but choosing a single operating point on that frontier is inherently application-dependent, and no universal criterion fixes the "right" level of sparsity, which trades support size against reconstruction fidelity. We adopt an elbow rule on the frontier as a sensible default, but the point it selects depends on the frontier's shape and on the smoothing used to locate the knee on it, both of which vary across data regimes and methods (Figure 4). Under severe contamination, for example, reconstruction error and support recovery anti-correlate, so a reconstruction-based elbow can settle at a too-dense operating point even though the frontier itself is well-behaved. In such regimes an alternative criterion—for instance one informed by the initialization-based stability of the selected support (Section 4.3)—may be preferable, and a selection rule that is robust across regimes is left to future work. The frontier construction itself is not the fragile part: because it is a continuous curve, refining the pruning-angle sweep moves the selected point only negligibly, and run-to-run variability is quantified by the stability analysis.

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

# A  Method Details

This appendix provides the full method specification referenced from Section 3.

## A.1  Scale Normalization and Dimensionless Scaling

Table 3: **Scale normalization and dimensionless scaling of objective terms.** Each term is written as an average over its natural index set. Additional normalizations make terms comparable across data sets and problem sizes.

| Term | What it measures | Scale normalization | Dimensionless / size-invariant scaling |
|---|---|---|---|
| Reconstruction $\mathcal{L}_{\text{recon}}(\mathbf{X}^*, \mathbf{W})$ | Residual energy on a mini-batch $\mathbf{X}^* \in \mathbb{R}^{b \times p}$, using raw $\mathbf{W}$ | Divide by robust variance $\hat{\sigma}_{\text{rob}}^2$ | Average per entry ($1/(bp)$) |
| Sparsity $\mathcal{L}_{\text{sparse}}(\mathbf{W})$ | Magnitude of individual loading entries, evaluated on a direction-only view of $\mathbf{W}$ | Column-direction normalization with gradient-detached norms | Average over parameters ($1/(pk)$); $\varepsilon$ set by Otsu's method |
| Orthogonality $\mathcal{L}_{\text{orth}}(\mathbf{W})$ | Correlation between components via Gram matrix of column-normalized loadings $\hat{\mathbf{W}}$ | Column normalization $\hat{\mathbf{W}}$ (gradients flow) | Average over Gram entries ($1/k^2$) |

Table 4: Scaling mechanisms ensuring each loss term is $O(1)$.

| Term | Mechanism | Result |
|---|---|---|
| $\mathcal{L}_{\text{recon}}$ | Divide by $\hat{\sigma}_{\text{rob}}^2$ | Scale-free; $O(1)$ for any data variance |
| $\mathcal{L}_{\text{sparse}}$ | $\log(1 + \hat{W}_{j\ell}^2/\varepsilon^2)$, mean over $pk$ | Bounded on $[0, 1]$ domain; $\varepsilon$ set by Otsu |
| $\mathcal{L}_{\text{orth}}$ | $\hat{\mathbf{W}}^\top \hat{\mathbf{W}}$ (unit columns), mean over $k^2$ entries | Gram entries in $[-1, 1]$; mean $O(1)$ |

In existing sparse PCA methods, the effective strength of the sparsity parameter changes with $p$, $k$, and feature variance, requiring re-tuning for each new data set. The normalization strategy above eliminates this dependence: a setting of $\lambda_s = 1.0$ means sparsity pressure is calibrated to the same magnitude as reconstruction pressure at PCA-equivalent quality, regardless of whether $p = 10$ or $p = 30{,}000$.

**PCA calibration computation.** The calibration constant $\mathcal{L}_{\text{recon}}^{\text{PCA}}$ is computed before training begins by evaluating the exact training loss function (including robust sample weighting and $\hat{\sigma}_{\text{rob}}^2$ normalization) at the PCA loading matrix $\mathbf{W}_{\text{PCA}}$. Specifically, $\mathbf{W}_{\text{PCA}}$ is obtained via randomized SVD (Halko et al., 2011) of the centered data (the same routine used for PCA initialization). $\mathcal{L}_{\text{recon}}^{\text{PCA}}$ is then averaged over $n_{\text{cal}}$ random minibatches (default $n_{\text{cal}} = 5$, each of size $b$) to reduce variance:

$$\mathcal{L}_{\text{recon}}^{\text{PCA}} = \frac{1}{n_{\text{cal}}} \sum_{m=1}^{n_{\text{cal}}} \mathcal{L}_{\text{recon}}(\mathbf{X}^{(m)}, \mathbf{W}_{\text{PCA}}), \tag{11}$$

where $\mathbf{X}^{(m)}$ denotes the $m$-th random minibatch. This is a property of the data and the number of components, not of the initialization strategy. When PCA components are provided externally, they are used directly. When they are not provided, the method computes them internally via randomized SVD. The cost is negligible relative to training (one SVD plus $n_{\text{cal}}$ forward passes).

## A.2 Robust Reconstruction Details

The robust variance estimate is $\hat{\sigma}_{\text{rob}}^2 = \left(\text{median}_j^+(\text{MAD}_j) \times 1.4826\right)^2$, where $\text{MAD}_j = \text{median}_i |x_{ij} - \bar{x}_j|$ is centered around the column mean (not the feature median), and $\text{median}_j^+$ denotes the median over features with $\text{MAD}_j > 0$. The 1.4826 factor is the Fisher consistency correction for Gaussian data.

Two design choices distinguish this estimator from the standard MAD. First, centering uses the column mean $\bar{x}_j$ rather than the per-feature median. This is necessary for conceptual consistency: the loss operates on centered data $\tilde{x}_{ij} = x_{ij} - \bar{x}_j$, so the scale estimate should reflect the spread of $\tilde{x}_{ij}$. For sparse count data (such as scRNA-seq) the feature median is typically zero while the feature mean is not, making the mean-centered MAD strictly more informative. Second, the outer median is restricted to features with $\text{MAD}_j > 0$. In high-dropout data, a majority of features may have $\text{MAD}_j = 0$, so the unrestricted median would also be zero. Restricting to nonzero MADs ensures $\hat{\sigma}_{\text{rob}}^2 > 0$ whenever at least one feature is expressed.

**Remark 1** (Robust weighting). *The* tanh *reweighting is differentiable everywhere and has bounded influence, so outlier resistance is built into the differentiable objective without introducing a non-differentiable cutoff.*

## A.3 Log Penalty Properties

The log penalty $\phi(x; \varepsilon) = \log(1 + x^2/\varepsilon^2)$ is a fully smooth ($C^\infty$) version of the log penalty $\log(|x| + \varepsilon)$ introduced by Candès et al. (2008). Its gradient structure provides intrinsic adaptive lasso behavior, as described in Section 3.1.3.

Near the origin, the Taylor expansion is $\phi(x) = x^2/\varepsilon^2 - x^4/(2\varepsilon^4) + O(x^6)$, showing that the penalty behaves as a scaled $\ell_2$ penalty for small entries. The second derivative is $\phi''(x) = 2(\varepsilon^2 - x^2)/(\varepsilon^2 + x^2)^2$, which is positive for $|x| < \varepsilon$ (locally convex) and negative for $|x| > \varepsilon$ (locally concave). The penalty is therefore convex near the origin and concave in the tails — sharing the concave-penalty structure of SCAD (Fan and Li, 2001) and MCP (Zhang, 2010), but without their non-differentiable points.

## A.4 Sparsity Warmup via Cosine Annealing

Applying full sparsity penalty from initialization can trap the optimizer in poor local minima. We gradually increase the sparsity weight using a cosine schedule:

$$\alpha(t) = \frac{1}{2}\left(1 - \cos\left(\frac{\pi \cdot t}{T_{\text{warm}}}\right)\right) \tag{12}$$

where $t$ is the current training step and $T_{\text{warm}}$ is the warmup period in steps.

## A.5 Orthogonality Penalty Details

We differentiate through the column normalization in $\hat{\mathbf{W}}$ (unlike the sparsity term, we do not detach column norms here). The unnormalized penalty $\|\mathbf{W}^\top\mathbf{W} - \mathbf{I}_k\|_F^2$ would simultaneously penalize deviations in column norm from 1 *and* inter-component correlation, creating a counterproductive tug-of-war with the sparsity penalty.

## A.6 Initialization

**PCA initialization (default).** The initial loading matrix consists of the top-$k$ right singular vectors of the centered data, computed via randomized SVD (Halko et al., 2011). For sparse or large-$p$ data, we use a `LinearOperator` wrapper that applies centering implicitly.

**Random orthonormal initialization.** When outlier contamination may have corrupted the covariance structure, we provide a covariance-independent alternative: (1) Draw $\mathbf{R} \in \mathbb{R}^{p \times k}$ with i.i.d. $\mathcal{N}(0, 1)$ entries. (2) QR-decompose to get orthonormal $\mathbf{Q}$. (3) Use $\mathbf{W}^{(0)} = \mathbf{Q}$ directly (unit-norm orthonormal columns, independent of the data scale).

## A.7 Optimization Details

Training is steps-based (not epoch-based). The learning rate follows a two-phase schedule: linear warmup from $\eta_{\max}/100$ to $\eta_{\max}$ over $T_{\text{lr-warm}} = T_{\text{warm}}/2$ steps, then cosine decay to $\eta_{\max}/10$. After the scheduled $T$ steps, training optionally continues at $\eta_{\max}/10$ with patience-based stopping (terminate when neither Gini nor reconstruction improves for $P$ consecutive checkpoints).

---

**Algorithm 1** DROSS-PCA Training Procedure

---

**Require:** Data $\mathbf{X} \in \mathbb{R}^{n \times p}$, components $k$, sparsity weight $\lambda_s$, orthogonality weight $\lambda_{\text{orth}}$, total steps $T$, warmup steps $T_{\text{warm}}$, $\varepsilon$-update interval $T_\varepsilon$, batch size $B$, patience $P$

1: **Preprocessing:**
2:    Compute global mean $\bar{\mathbf{x}} = \frac{1}{n} \sum_i \mathbf{x}_i$
3:    Compute per-feature MAD: $\text{MAD}_j = \text{median}_i |x_{ij} - \bar{x}_j|$
4:    Compute robust scale $\hat{\sigma}_{\text{rob}}^2 = (\text{median}_j^+ \text{MAD}_j \times 1.4826)^2$
5:    Compute $\mathcal{L}_{\text{recon}}^{\text{PCA}}$ from PCA components
6: **Initialization:**
7:    $\mathbf{W}^{(0)} \leftarrow$ PCA (top-$k$ right singular vectors) *or* random orthonormal $\mathbf{Q}$
8:    $\varepsilon \leftarrow 1/\sqrt{p}$
9: **for** $t = 1, 2, \dots$ **do**
10:     Sample minibatch $\mathbf{X}_b$ of size $B$ from infinite stream
11:     $\alpha \leftarrow \frac{1}{2}(1 - \cos(\pi t / T_{\text{warm}}))$
12:     Compute $\nabla_{\mathbf{W}} \mathcal{L}$ via autodifferentiation
13:     Update $\mathbf{W}$ using AdamW with scheduled learning rate
14:     **if** $t \bmod T_\varepsilon = 0$ **and** $t \geq T_{\text{warm}}$ **then**
15:         $\varepsilon \leftarrow \max(\text{Otsu}(|\hat{\mathbf{W}}|), 1/\sqrt{p})$
16:     **end if**
17:     **if** $t = T$ **then**
18:         Switch to constant LR $= \eta_{\max}/10$; begin patience tracking
19:     **end if**
20:     **if** $t > T$ **and** no improvement for $P$ checkpoints **then**
21:         **break**
22:     **end if**
23: **end for** **return** $\mathbf{W}^{(t)}$

---

## A.8 Cosine-Preserving Pruning Details

CPP prunes entries in rank order of $|w_i|$ (smallest first). Because $w_{\text{cpp}}$ is obtained by zeroing a subset of entries, $\cos(w, w_{\text{cpp}}) = \|w_{\text{cpp}}\|_2 / \|w\|_2$, so the cosine constraint is equivalent to preserving at least $\cos_t^2$ of the squared $\ell_2$ energy. We define effective sparsity as:

$$\text{EffSparse}_{\text{cpp}}(W; \cos_t) = \frac{1}{pk} \sum_{j=1}^{p} \sum_{\ell=1}^{k} \mathbf{1}\left\{ (W_{\text{cpp}}(\cos_t))_{j\ell} = 0 \right\}. \tag{13}$$

---

**Algorithm 2** Cosine-Preserving Pruning (CPP)

---

**Require:** Trained loading matrix $\mathbf{W} \in \mathbb{R}^{k \times p}$, cosine threshold $\cos_t \in (0, 1]$
1: **for** each component $j = 1, \ldots, k$ **do**
2:     $E_j \leftarrow \|\mathbf{w}_j\|_2^2$                                                        ▷ total energy
3:     $\pi_j \leftarrow \operatorname{argsort}(|w_{j,1}|, \ldots, |w_{j,p}|)$                             ▷ ascending
4:     $C_j[i] \leftarrow \sum_{m=1}^{i} w_{j,\pi_j[m]}^2$ for $i = 1, \ldots, p$            ▷ cumulative removed energy
5: **end for**

6: **for** each component $j = 1, \ldots, k$ **do**
7:     $E_{\min} \leftarrow \cos_t^2 \cdot E_j$                                        ▷ energy to preserve
8:     Find largest $i^*$ s.t. $E_j - C_j[i^*] \geq E_{\min}$                     ▷ binary search
9:     $w_{j,\pi_j[1:i^*]} \leftarrow 0$                              ▷ zero the $i^*$ smallest entries
10: **end forreturn** Pruned $\mathbf{W}$

---

**Operating point selection.** For each model, we select a single operating point from the CPP sparsity–VE frontier. Both axes are normalized to $[0, 1]$ per curve and the VE curve is smoothed via a Savitzky–Golay filter (Savitzky and Golay, 1964). We compute the rate of change of the tangent angle $\vartheta = \arctan(d\,\text{VE}/d\sigma)$ with respect to normalized sparsity $\sigma$:

$$\frac{d\vartheta}{d\sigma} = \frac{d^2\text{VE}/d\sigma^2}{1 + (d\,\text{VE}/d\sigma)^2}. \tag{14}$$

The operating point is the sparsity at which $d\vartheta/d\sigma$ is most negative — the sharpest transition from flat to steep on the frontier. For hyperparameter selection across configurations, we apply the same criterion to the frontier of best-per-configuration operating points.

## B Simulation Details

### B.1 Simulation Designs

Sparse PCA serves two distinct purposes: dimensionality reduction and feature selection. These are not the same objective. Moreover, individual sparse components may not be identifiable when leading eigenvalues are close (Ma, 2013), and whether the "correct" set of active features is well-defined depends on identifiability conditions (Lei and Vu, 2015). We organize the evaluation around three questions: (1) sparsity–reconstruction tradeoff, (2) feature identification, and (3) stability across initializations.

**Spiked covariance model.** The spiked covariance model and the generating-basis terminology are introduced in Section 4.1 (Eq. 10). We elaborate here on the relationship between the generating basis $\mathbf{V}$ and the eigenbasis of $\Sigma$.

When the columns of $\mathbf{V}$ have disjoint support (S1, S2), $\mathbf{V}^\top \mathbf{V} = \mathbf{I}_k$ and the generating basis coincides with an eigenbasis of $\Sigma$. When columns overlap (S3, S4), $\mathbf{V}^\top \mathbf{V} \neq \mathbf{I}_k$, so $\mathbf{VDV}^\top$ is not an eigendecomposition and the eigenvectors of $\Sigma$ are rotations of the columns of $\mathbf{V}$. In either case, the column space of $\mathbf{V}$ equals the top-$k$ eigenspace of $\Sigma$.

This distinction clarifies which comparisons are well-posed under each identifiability regime. The column space of $\mathbf{V}$ and the generating support are always meaningful targets. Comparisons against individual generating components are informative only when the generating basis is identifiable: under eigenvalue degeneracy (S2), any rotation within the eigenspace is equally optimal, so different bases may have different per-column support patterns. Under overlap (S3, S4), the generating components are not eigenvectors of $\Sigma$.

Crossing support overlap (0% vs. 25%) with eigenvalue separation gives four configurations (S1–S4). Details on the overlap structure, outlier contamination, and replication strategy are as described in the main text.

## B.2 Evaluation Metrics

We evaluate each method on both model-free metrics (sparsity, variance explained, orthogonality) and metrics that compare against the generating basis (global AUPRC, per-column AUPRC). The AUPRC metrics require precise specification because they involve choices about scoring functions and component matching.

**Global AUPRC.** Global AUPRC measures whether the method identifies the correct set of active features, without regard to which component each feature belongs to. Given estimated loadings $\hat{\mathbf{W}} \in \mathbb{R}^{p \times k}$ and a binary indicator $y_j \in \{0, 1\}$ for whether feature $j$ is in the generating support:

1. Compute the feature importance score as the row norm of the estimated loading matrix: $s_j = \|\hat{\mathbf{w}}_{j \cdot}\|_2 = \sqrt{\sum_{\ell=1}^{k} \hat{W}_{j\ell}^2}$ for $j = 1, \dots, p$.

2. Compute the area under the precision–recall curve (AUPRC) using $s_j$ as the predicted score and $y_j$ as the binary label, via `sklearn.metrics.average_precision_score`.

Global AUPRC $= 1.0$ means the row norms perfectly separate active from inactive features at some threshold. It is invariant to component permutation and does not require matching estimated components to generating components.

**Per-column AUPRC.** Per-column AUPRC measures whether each estimated component loads on the correct features, requiring a correspondence between estimated and generating components. Because the component ordering is arbitrary, we establish this correspondence via optimal assignment (Hungarian algorithm) on AUPRC scores.

Given estimated loadings $\hat{\mathbf{W}} \in \mathbb{R}^{p \times k}$ and per-column generating supports $S_1, \dots, S_{k_{\text{gen}}}$ (each $S_j$ is a set of feature indices):

1. Compute the $k \times k_{\text{gen}}$ AUPRC matrix $A$, where entry $A_{i,j}$ is the AUPRC obtained by using $|\hat{W}_{\cdot,i}|$ (absolute values of the $i$-th estimated column) as the predicted score and the binary indicator of $S_j$ as the label.

2. Match estimated components to generating components via optimal assignment: solve for the permutation $\pi$ that maximizes $\sum_i A_{i,\pi(i)}$ using the Hungarian algorithm (`scipy.optimize.linear_sum_assignment`).

3. Per-column AUPRC is the mean of the $\min(k, k_{\text{gen}})$ matched AUPRC values.

Per-column AUPRC $= 1.0$ means each estimated component perfectly separates the features in its matched generating support from all other features. Unlike global AUPRC, it is sensitive to whether the method assigns features to the *correct* generating component, not just whether it identifies them as active. This metric is most informative when the generating basis is identifiable (S1, S1c). Under eigenvalue degeneracy (S2) or overlapping support (S3, S4), per-column AUPRC conflates identifiability limitations with estimation quality. See Section 4.1 for discussion.

**Subspace angles.** Principal angles between the column spaces of the estimated loadings and the generating basis measure subspace recovery independently of basis choice. Given $\hat{\mathbf{W}} \in \mathbb{R}^{p \times k}$ and $\mathbf{V} \in \mathbb{R}^{p \times k}$, the $k$ principal angles $\theta_1 \leq \cdots \leq \theta_k$ are computed via `scipy.linalg.subspace_angles`. We report $\overline{\sin \theta} = \frac{1}{k} \sum_{\ell=1}^{k} \sin \theta_\ell$, which equals 0 when the subspaces are identical and approaches 1 for maximally misaligned subspaces. Because subspace angles depend only on the column spans of $\hat{\mathbf{W}}$ and $\mathbf{V}$, they are invariant to rotation within each subspace. This makes them the appropriate recovery metric under eigenvalue degeneracy (S2), where the generating basis is non-identifiable but its column space is.

**Orthogonality.** We measure inter-component correlation via the mean absolute off-diagonal entry of the direction-normalized Gram matrix. Given $\hat{\mathbf{W}} \in \mathbb{R}^{p \times k}$:

1. Normalize each column to unit norm: $\tilde{\mathbf{w}}_\ell = \hat{\mathbf{w}}_\ell / \|\hat{\mathbf{w}}_\ell\|_2$.

2. Compute the Gram matrix $G = \tilde{\mathbf{W}}^\top \tilde{\mathbf{W}}$.

3. Orthogonality $= \frac{1}{k(k-1)} \sum_{i \neq j} |G_{ij}|$.

Orthogonality $= 0$ for perfectly orthogonal components. It approaches 1 for identical components. This metric is computed on the direction-normalized loadings, so it measures angular relationships between components independent of column magnitudes. It is consistent across methods regardless of how zeros were produced ($\ell_1$ thresholding, CPP pruning, or native).

**Additional metrics.** *Variance explained:* VE $= 1 - \|\tilde{\mathbf{X}} - \tilde{\mathbf{X}}\hat{\mathbf{W}}\hat{\mathbf{W}}^\top\|_F^2 / \|\tilde{\mathbf{X}}\|_F^2$, where $\tilde{\mathbf{X}}$ is centered data. *Sparsity:* fraction of exactly zero entries in $\hat{\mathbf{W}}$. *Relative reconstruction error:* $\|\tilde{\mathbf{X}} - \tilde{\mathbf{X}}\hat{\mathbf{W}}\hat{\mathbf{W}}^\top\|_F / \|\tilde{\mathbf{X}}\|_F$.

### B.3 Baselines and Configuration

We compare against four sparse PCA methods plus standard PCA, all with PCA initialization.

**Standard PCA.** Randomized SVD (Halko et al., 2011). 1 configuration.

**Zou–Hastie SparsePCA (Zou et al., 2006).** $\alpha \in \{0.01, 0.05, 0.1, 0.2, 0.5, 1.0, 2.0, 5.0\}$, ridge $= 0.01$. 8 configurations.

**MiniBatch SparsePCA (Pedregosa et al., 2011).** Same $\alpha$ grid. 8 configurations.

**VarPro SPCA (Erichson et al., 2020).** $\alpha \in \{10^{-4}, \dots, 25\}$, $\beta = 0.001$. 13 configurations. Its robust variant adds a sparse error term over individual entries (a robust-PCA-style decomposition), targeting cellwise rather than the casewise contamination of our S1c benchmark, so we run the standard variant.

**PMD/SPC (Witten et al., 2009).** $\|v\|_1 \leq c$, $c = 1 + f(\sqrt{p} - 1)$ with $f \in \{0.05, \dots, 0.8\}$. 8 configurations.

**DROSS-PCA.** $\lambda_s \in \{0, 0.001, 0.01, 0.1, 0.5, 1.0\} \times \lambda_{\mathrm{orth}} \in \{0, 0.001, 0.01, 0.1, 0.5, 1.0\}$. 36 configurations. Fixed parameters in Table 5.

**Two-level operating point selection.** Each method sweeps a grid of hyperparameter configurations, and each configuration produces a model at a particular point on the sparsity–reconstruction frontier. Selecting a single representative result per method and replicate requires two levels of selection for DROSS-PCA and one level for baselines.

*Level 1 (within-model, DROSS-PCA only).* Each trained DROSS-PCA model is followed by a 100-point CPP sweep over $\theta \in [0°, 10°]$, tracing a sparsity–VE frontier. The operating point is selected via the $d\vartheta/d\sigma$ criterion (Section 3.2), which identifies the knee of the (sparsity, variance explained) curve. This produces one (sparsity, reconstruction error) pair per hyperparameter configuration. For baselines, each configuration directly produces native sparsity with no sweep.

*Level 2 (across configurations, all methods).* Given the set of (sparsity, relative reconstruction error) pairs across all hyperparameter configurations for a method, we select the best configuration via the chord-elbow method on the (sparsity, relative reconstruction error) frontier. Both axes are normalized to $[0, 1]$, points are sorted by sparsity, and the selected configuration is the one with maximum perpendicular distance from the chord connecting the least-sparse and most-sparse points. This selects the configuration that achieves the best tradeoff between sparsity and reconstruction quality.

Table 5: Factorial simulation design. The spiked covariance model (Johnstone and Lu, 2009) with $p = 500$, $k = 5$, $n = 1,000$, $s = 30$ nonzero entries per component. Support overlap and eigenvalue separation are crossed to produce four structural configurations (S1–S4), each evaluated at three SNR levels with 100 replicates per cell.

| | Disjoint support (0%) | | Overlapping support (25%) | |
|---|---|---|---|---|
| | Separated eig. | Similar eig. | Separated eig. | Similar eig. |
| Configuration | S1 | S2 | S3 | S4 |
| Spike eigenvalues | $10 \rightarrow 5$ | all $= 7.5$ | $10 \rightarrow 5$ | all $= 7.5$ |
| Identifiability | Full | Eig. degeneracy | Support ambiguity | Max. ambiguous |
| *SNR levels (crossed with each configuration)* | | | | |
| Low SNR | $\sigma^2 = 0.675$ | (VE $\approx 10\%$) | | |
| Medium SNR | $\sigma^2 = 0.075$ | (VE $\approx 50\%$) | | |
| High SNR | $\sigma^2 = 0.0083$ | (VE $\approx 90\%$) | | |
| *Methods and hyperparameter grids (each run on all 1,200 data sets)* | | | | |
| Standard PCA | Randomized SVD, 1 configuration | | | |
| Zou–Hastie | $\alpha \in \{0.01, 0.05, 0.1, 0.2, 0.5, 1.0, 2.0, 5.0\}$, ridge $= 0.01$ (8 configs) | | | |
| MiniBatch | $\alpha \in \{0.01, 0.05, 0.1, 0.2, 0.5, 1.0, 2.0, 5.0\}$, ridge $= 0.01$ (8 configs) | | | |
| VarPro | $\alpha \in \{10^{-4}, \dots, 25\}$, $\beta = 0.001$ (13 configs) | | | |
| PMD/SPC | $\|v\|_1 \leq 1 + f(\sqrt{p} - 1)$, $f \in \{0.05, \dots, 0.8\}$ (8 configs) | | | |
| DROSS-PCA | $\lambda_s \in \{0, 0.001, 0.01, 0.1, 0.5, 1\} \times \lambda_{\mathrm{orth}} \in \{0, 0.001, 0.01, 0.1, 0.5, 1\}$ (36 configs) | | | |
| Data Sets | 4 configs $\times$ 3 SNR $\times$ 100 replicates $= 1,200$ | | | |
| Total fits | $1,200 \times 74$ method configs $= 88,800$ | | | |

DROSS-PCA fixed parameters: PCA initialization, $T = 3,000$ steps, $T_{\mathrm{warm}} = 1,000$, $\eta_{\max} = 0.01$, $b = 1,000$, patience $= 5$. Each DROSS-PCA model is evaluated on a 100-point CPP sweep ($\theta \in [0°, 10°]$).

Table 6: Outlier contamination experiment. S1c crosses contamination fraction $\{2\%, 5\%\}$ with outlier magnitude $\{4\times, 8\times\}$ the data's standard deviation, applied to S1 (disjoint support, separated eigenvalues). This isolates the contribution of robust sample weighting from overlap and degeneracy effects.

| Parameter | Value |
|---|---|
| Base structure | S1: disjoint support, separated eigenvalues |
| Dimensions | $p = 500$, $k = 5$, $n = 1,000$, $s = 30$ |
| Contamination fraction | $\{2\%, 5\%\}$ of samples replaced |
| Outlier magnitude | $\{4\times, 8\times\}$ the data's std |
| SNR levels | Low, Medium, High (same $\sigma^2$ as factorial) |
| Replicates | 100 per configuration |
| Methods | Same 6 methods and grids as factorial (74 configs) |
| Data Sets | 4 configs $\times$ 3 SNR $\times$ 100 replicates $= 1,200$ |
| Total fits | $1,200 \times 74 = 88,800$ |

## C  Per-Configuration Results

This appendix provides the full VE–sparsity and recovery figures for each factorial configuration, supplementing the synthesized results in Section 4.1.

**S1: Disjoint support, separated eigenvalues (Figure 2, 10).** With disjoint support and separated eigenvalues, all methods perform well. At medium and high SNR, Zou–Hastie, VarPro, and DROSS-PCA all achieve $> 90\%$ sparsity with variance explained within 1% of PCA. DROSS-PCA achieves per-column AUPRC 0.927 at high SNR with orthogonality 0.003. MiniBatch SparsePCA achieves comparable sparsity–VE tradeoffs but with elevated orthogonality (0.075, versus $\leq 0.01$ for all other methods), indicating that its components are poorly separated despite adequate reconstruction and sparsity.

**S2: Eigenvalue degeneracy (Figures 11, 12).** With degenerate eigenvalues, the component basis is non-identifiable: any rotation within the eigenspace is equally optimal. Subspace angles confirm that

Table 7: Identifiability analysis across random initializations. DROSS-PCA is run with 100 independent random orthonormal initializations per replicate at fixed hyperparameters, across all four factorial configurations to provide identifiable controls (S1, S2) and overlap conditions (S3, S4).

| Parameter | Value |
|---|---|
| Configurations | S1 (disjoint, separated), S2 (disjoint, similar), |
| | S3 (overlap, separated), S4 (overlap, similar) |
| SNR levels | Low, Medium, High |
| Replicates | 100 per configuration $\times$ SNR |
| Initializations | 100 random orthonormal seeds per replicate |
| Fixed hyperparameters | $\lambda_s = 1.0$, $\lambda_{\mathrm{orth}} = 1.0$ |
| Training | $T = 3{,}000$ steps, $T_{\mathrm{warm}} = 1{,}000$, $\eta_{\max} = 0.01$, $b = 1{,}000$, patience $= 5$ |
| Data Sets | 4 configs $\times$ 3 SNR $\times$ 100 replicates $= 1{,}200$ |
| Total fits | $1{,}200 \times 100$ seeds $= 120{,}000$ |

Each of the 120,000 models is evaluated on a 100-point CPP sweep. Identifiability metrics: Q1–Q4 (Section 4.3).

Table 8: Feature identifiability across SNR levels (mean $\pm$ std over 100 replicates, 100 random initializations each). Global AUPRC measures generating support recovery. Q2 and Q3 measure per-feature assignment consistency. Pairwise false co-assignment (Q4 not) increases with SNR under overlap, as stronger signal makes shared structure more visible. All entries were recomputed after the revision to the robust reweighting step. The reported patterns and conclusions are unchanged.

| Config | SNR | Global AUPRC | Q1 (active) | Q2 (private) | Q3 (shared) | Q4 co (same) | Q4 not (diff) |
|---|---|---|---|---|---|---|---|
| | Low | $.847 \pm .051$ | $.782 \pm .073$ | $.778 \pm .072$ | — | $.616 \pm .097$ | $.002 \pm .004$ |
| S1 | Med | $.927 \pm .021$ | $.895 \pm .030$ | $.891 \pm .029$ | — | $.801 \pm .054$ | $.002 \pm .003$ |
| | High | $.966 \pm .014$ | $.952 \pm .020$ | $.948 \pm .020$ | — | $.905 \pm .038$ | $.002 \pm .003$ |
| | Low | $.871 \pm .048$ | $.816 \pm .068$ | $.805 \pm .063$ | — | $.666 \pm .100$ | $.005 \pm .008$ |
| S2 | Med | $.954 \pm .019$ | $.934 \pm .028$ | $.929 \pm .026$ | — | $.872 \pm .051$ | $.003 \pm .003$ |
| | High | $.979 \pm .015$ | $.969 \pm .021$ | $.963 \pm .020$ | — | $.940 \pm .041$ | $.003 \pm .007$ |
| | Low | $.875 \pm .047$ | $.838 \pm .061$ | $.744 \pm .072$ | $.617 \pm .116$ | $.634 \pm .087$ | $.023 \pm .023$ |
| S3 | Med | $.940 \pm .023$ | $.922 \pm .030$ | $.792 \pm .075$ | $.805 \pm .082$ | $.807 \pm .058$ | $.060 \pm .053$ |
| | High | $.965 \pm .020$ | $.955 \pm .026$ | $.715 \pm .110$ | $.890 \pm .074$ | $.887 \pm .056$ | $.139 \pm .086$ |
| | Low | $.880 \pm .042$ | $.845 \pm .055$ | $.737 \pm .068$ | $.636 \pm .113$ | $.647 \pm .092$ | $.031 \pm .029$ |
| S4 | Med | $.950 \pm .023$ | $.936 \pm .030$ | $.777 \pm .081$ | $.840 \pm .089$ | $.839 \pm .064$ | $.081 \pm .061$ |
| | High | $.963 \pm .021$ | $.952 \pm .027$ | $.726 \pm .117$ | $.892 \pm .069$ | $.883 \pm .056$ | $.134 \pm .098$ |

DROSS-PCA recovers the subspace as well as Zou–Hastie (mean $\sin\theta = 0.018$ vs. $0.017$ at high SNR). The per-column AUPRC gap (0.883 vs. 0.978) reflects basis rotation, not estimation failure. DROSS-PCA maintains the lowest orthogonality among high-sparsity methods (0.002).

**S3: Overlapping support (Figures 13, 14).** DROSS-PCA achieves VE $= 0.901$ at high SNR (99.9% of PCA), compared to 0.888 for Zou–Hastie. Zou–Hastie's orthogonality is 0.033 (absent in S1/S2), while DROSS-PCA maintains 0.007–0.013. VarPro's per-column AUPRC collapses from 0.907 (medium SNR) to 0.750 (high SNR), likely due to its variable-projection formulation struggling with shared features at high signal strength.

**S4: Maximally ambiguous (Figures 15, 16).** DROSS-PCA achieves VE $= 0.901$ at high SNR with 91.6% sparsity, global AUPRC 0.978, and orthogonality 0.008. Zou–Hastie achieves higher per-column AUPRC (0.967) but with $4\times$ worse orthogonality (0.033).

**S1c: Outlier contamination (Figures 3, 17–20).** The contamination study crosses fraction $\{2\%, 5\%\}$ with magnitude $\{4\times, 8\times\}$. At mild contamination (2% at $4\times$), DROSS-PCA and the non-robust baselines achieve global AUPRC $\sim 0.92$–$0.98$ at medium and high SNR. At the most severe condition (5% at $8\times$), DROSS-PCA achieves 0.74 at low SNR while the non-robust baselines remain near $\sim 0.32$. At medium SNR,

Table 9: Computational scaling experiments. Problem size increases jointly in $p$, $n$, and $k$ with $s = 30$ held fixed, so per-component sparsity structure is constant across the sweep. The ratio $n/p = 10$ is fixed throughout. The robust methods (ROSPCA, sPCAgrid) and the CPU-bound baselines (Zou–Hastie, MiniBatch, VarPro, PMD) are skipped where computationally infeasible. DROSS-PCA is additionally timed on GPU. Structure: S1 (disjoint, separated), $\sigma^2 = 1.0$, 1 replicate per size.

| $p$ | $n$ | $k$ | $s$ | Methods run | Configs |
|---|---|---|---|---|---|
| 500 | 5,000 | 10 | 30 | All 8 | 90 |
| 1,000 | 10,000 | 20 | 30 | All 8 | 90 |
| 2,000 | 20,000 | 50 | 30 | All 8 | 90 |
| 5,000 | 50,000 | 100 | 30 | All 8 | 90 |
| 10,000 | 100,000 | 200 | 30 | All 8 | 90 |
| Total fits | | | | | 450 |

Same hyperparameter grids as factorial. Wall-clock time and peak memory recorded for each run.

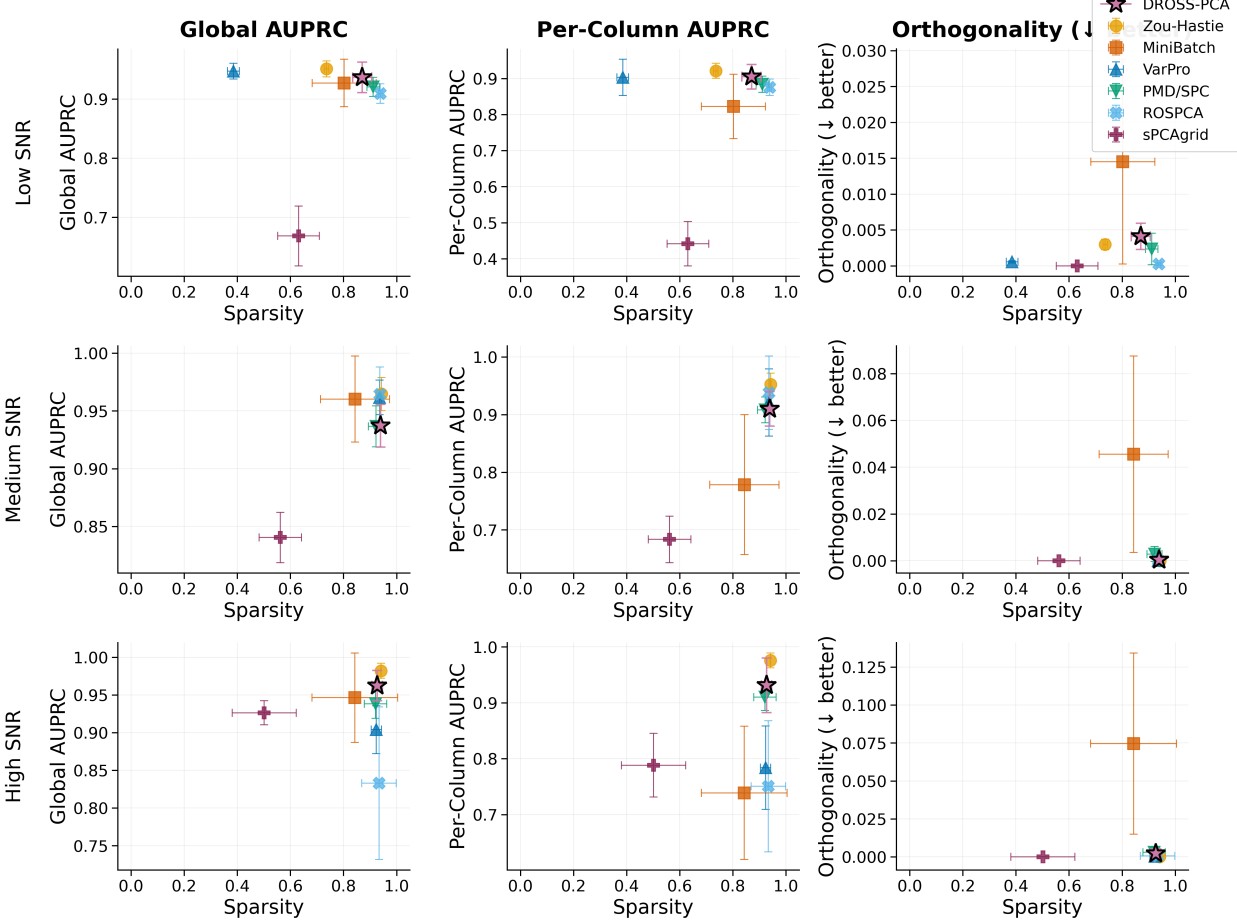

Figure 10: Feature recovery metrics for S1. Per-column AUPRC is the appropriate secondary metric (basis identifiable).

DROSS-PCA achieves 0.92 versus the best non-robust baseline at ∼0.82 (PMD/SPC), while the robust ROSPCA reaches 0.97. The advantage over non-robust methods emerges progressively as contamination severity increases.

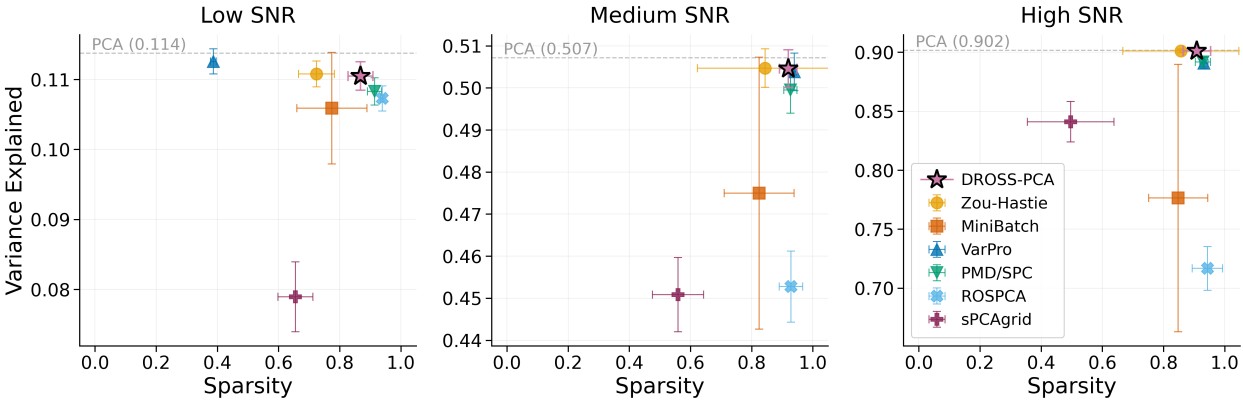

Figure 11: Sparsity vs. variance explained for S2.

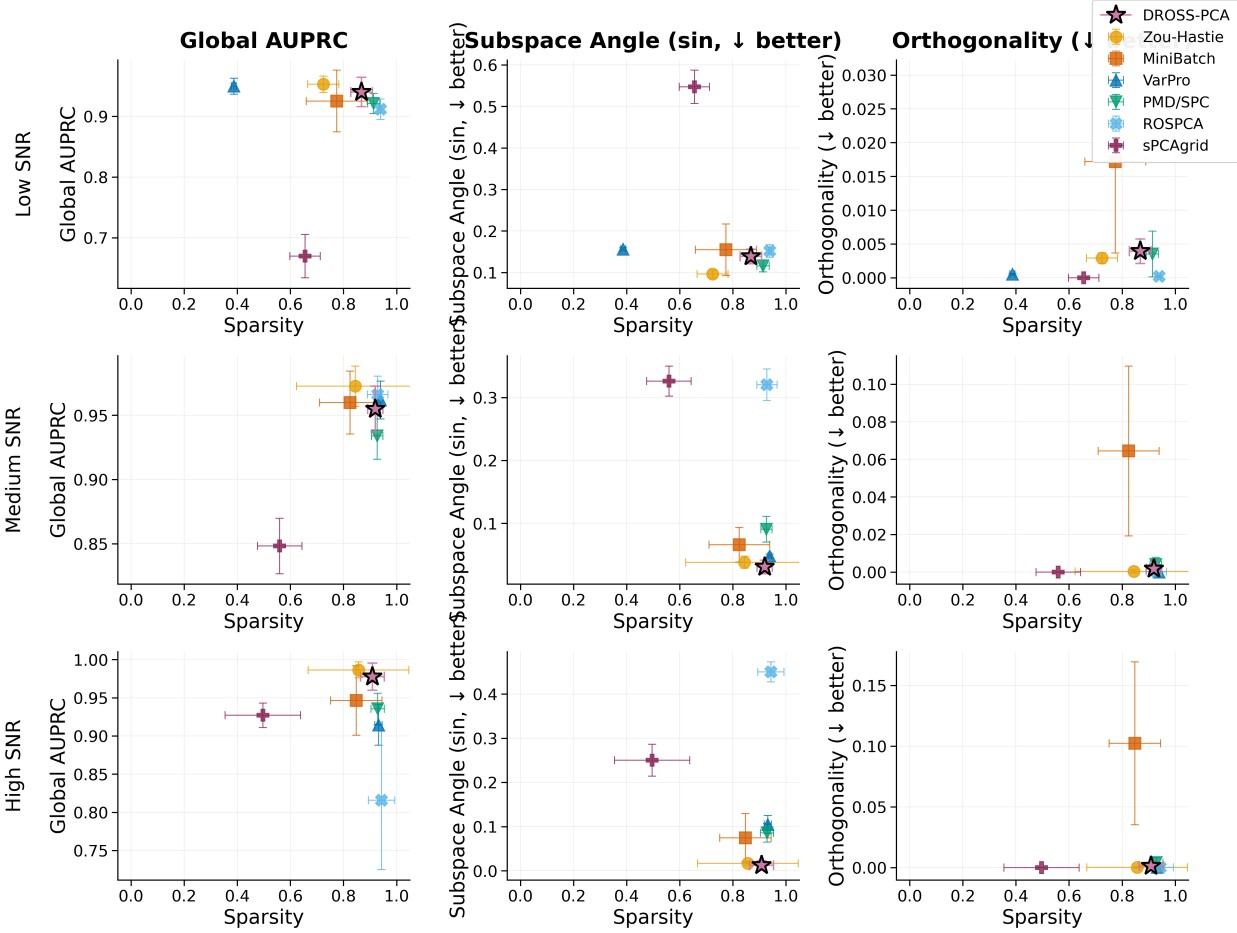

Figure 12: Feature recovery metrics for S2. Subspace angle (center) is the appropriate secondary metric (basis non-identifiable under eigenvalue degeneracy).

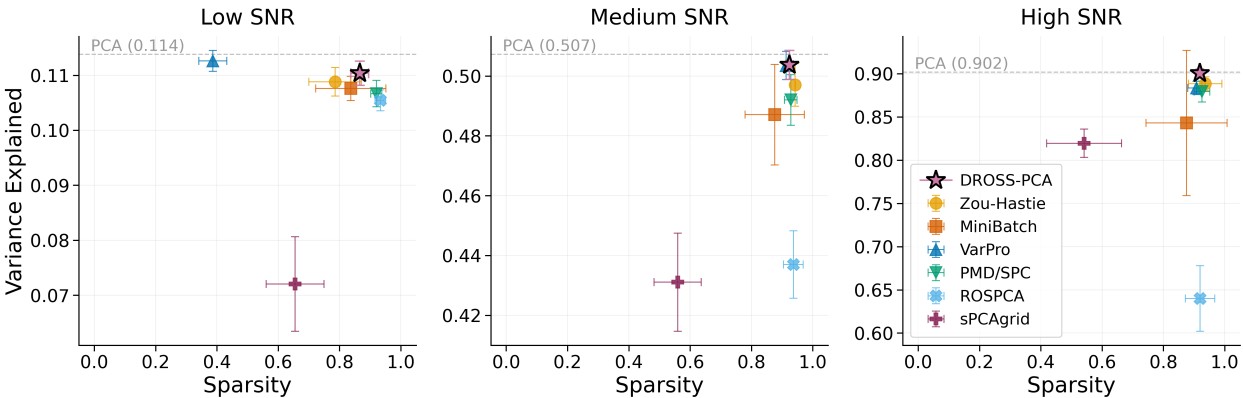

Figure 13: Sparsity vs. variance explained for S3.

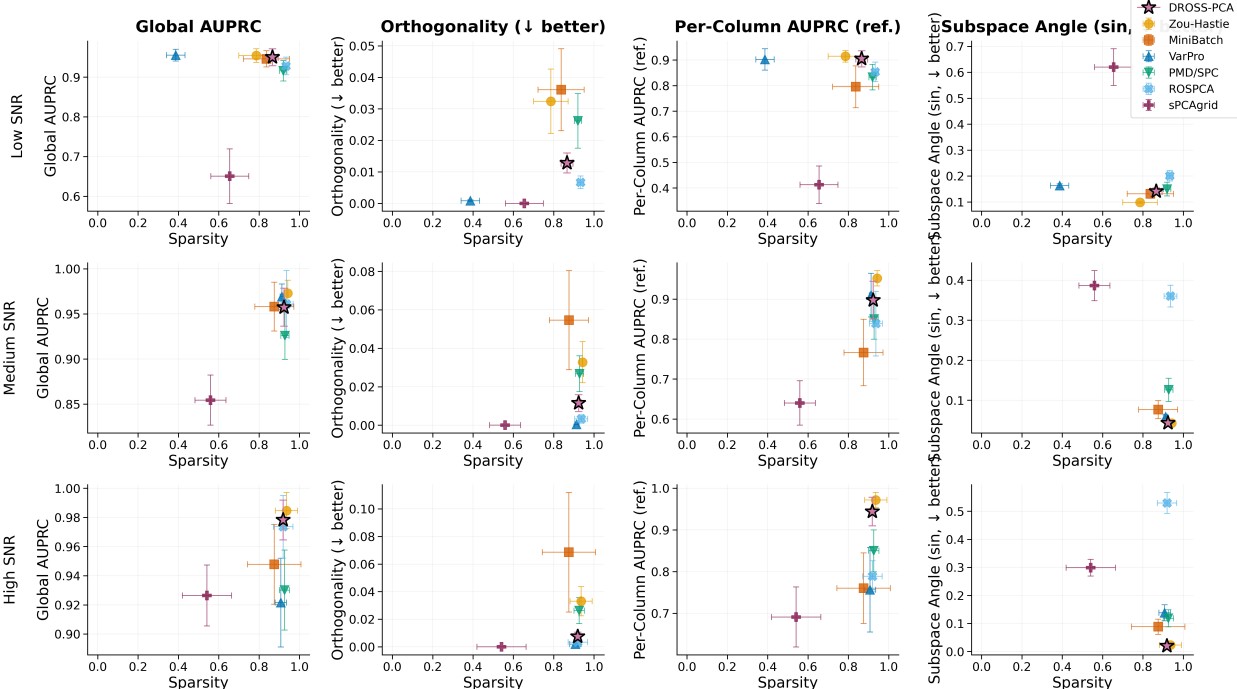

Figure 14: Feature recovery metrics for S3. Orthogonality (center) is the key diagnostic under overlapping support.

## D   Application Details: Human Lung Cell Atlas

**Data.**   The HLCA core (Sikkema et al., 2023) contains 584,944 cells and 27,402 genes after standard preprocessing (log-normalization). The data matrix is stored as a sparse CSR matrix ($\sim$3 GB, 1.14 billion nonzero entries). The dense representation would require $\sim$64 GB in float32.

**PCA.**   GPU-accelerated PCA ($k = 5{,}000$) is computed via rapids-singlecell on the sparse matrix. This provides the calibration constant $\mathcal{L}_{\text{recon}}^{\text{PCA}}$ and the variance explained reference ($\sim$81% at $k = 5{,}000$).

**Baselines.**   All existing sparse PCA implementations (Zou–Hastie, MiniBatch, VarPro, PMD) require the dense data matrix in memory. At $584{,}944 \times 27{,}402$ in float64, this is $\sim$128 GB — exceeding typical single-

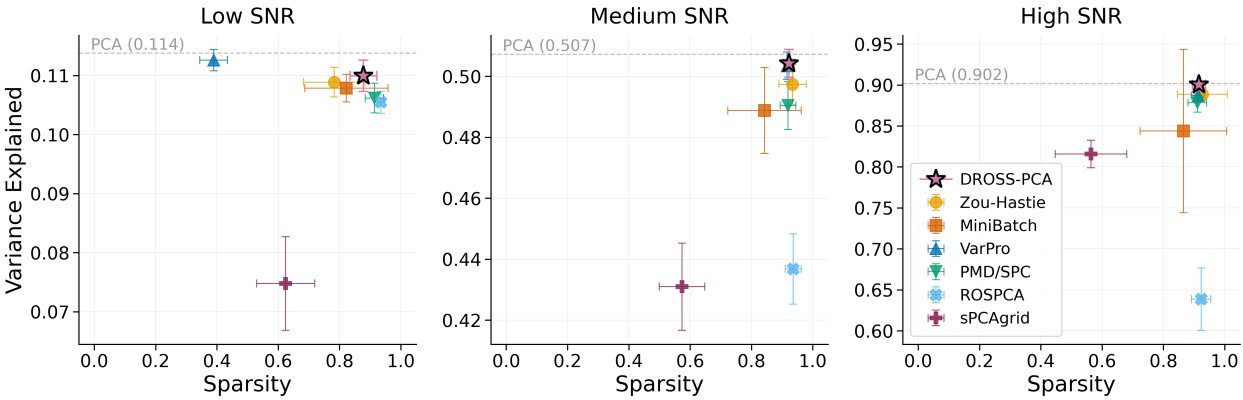

Figure 15: Sparsity vs. variance explained for S4.

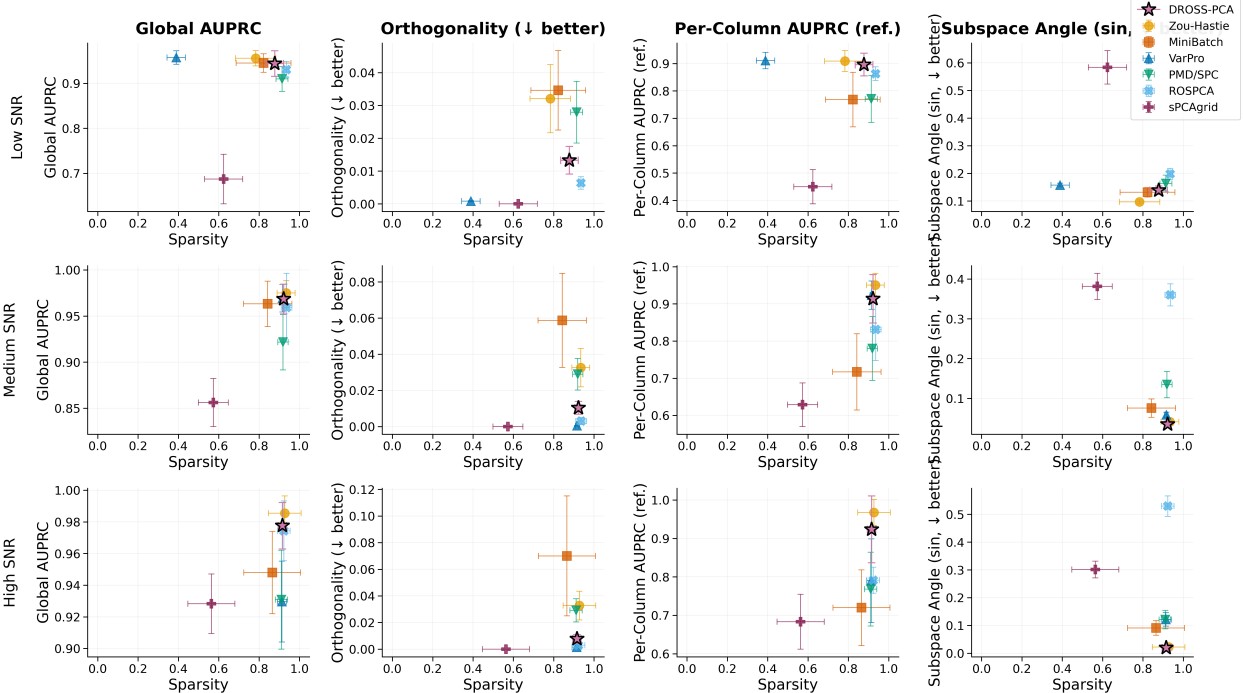

Figure 16: Feature recovery metrics for S4.

machine RAM. DROSS-PCA operates directly on the sparse representation, densifying only one minibatch at a time $(10{,}000 \times 27{,}402 \times 8 \approx 2.2\,\mathrm{GB})$.

**Training configuration.** $\lambda_s = 0.5$, $\lambda_{\mathrm{orth}} = 0.5$, $T = 3{,}000$ steps, $T_{\mathrm{warm}} = 1{,}000$, $\eta_{\mathrm{max}} = 5 \times 10^{-4}$, batch size $= 10{,}000$, patience $= 5$. We run 100 independent random orthonormal initializations at $k = 5{,}000$.

**Stability analysis pipeline.** Each of the 100 models is followed by a 100-point CPP sweep ($\theta_{\mathrm{max}} = 10°$, float32 precision). The operating point is selected via the $d\theta/ds$ criterion (Section 3.2). Pruned loadings are saved as sparse CSR matrices.

**Consensus core computation.** Given 100 pruned loading matrices $\mathbf{W}^{(1)}, \ldots, \mathbf{W}^{(100)}$, each of shape $(k, p)$ with $k = 5{,}000$ and $p = 27{,}402$, we compute consensus cores as follows.

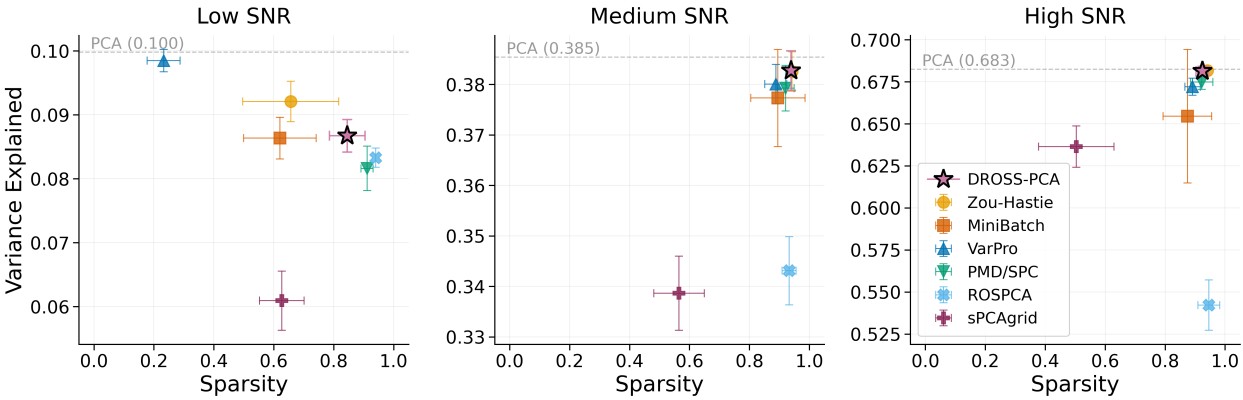

Figure 17: Sparsity vs. variance explained: S1c, 2% at 4×.

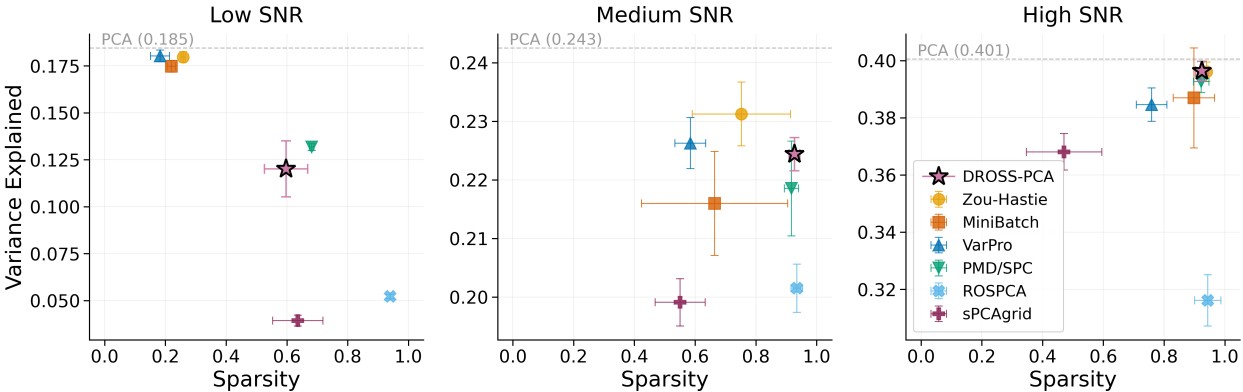

Figure 18: Sparsity vs. variance explained: S1c, 2% at 8×.

*Step 1: Binarize.* For each run $r$, construct a binary matrix $B^{(r)} \in \{0,1\}^{k \times p}$ where $B_{jg}^{(r)} = 1$ if gene $g$ has a nonzero loading in component $j$.

*Step 2: Jaccard matching.* Select one run as the reference (run $r = 0$). For each other run $r'$, compute the $k \times k$ Jaccard similarity matrix between all pairs of components: $J_{ij} = |S_i^{(0)} \cap S_j^{(r')}| / |S_i^{(0)} \cup S_j^{(r')}|$, where $S_j^{(r)}$ is the support (nonzero gene set) of component $j$ in run $r$. This is computed efficiently via GPU matrix multiplication: $B^{(0)} B^{(r')\top}$ gives intersection sizes, from which Jaccard follows. Apply the Hungarian algorithm (Kuhn, 1955) to $-J$ to obtain an optimal one-to-one assignment $\sigma_{r'} : [k] \to [k]$ maximizing total Jaccard. This ensures no two reference components map to the same target, preventing inflated overlap scores.

*Step 3: Gene frequency.* A reference component $j$ counts as matched in run $r'$ when its assigned partner clears the acceptance threshold $J_{j,\sigma_{r'}(j)} \geq 0.1$. Let $n_j$ be the number of the 99 other runs in which $j$ is matched (median $n_j = 99$ across the 5,000 components). The *gene frequency* $f_{jg}$ is the fraction of $j$'s matched partners in which gene $g$ is active: $f_{jg} = \frac{1}{n_j} \sum_{r': \text{matched}} B_{\sigma_{r'}(j), g}^{(r')}$.

*Step 4: Consensus core.* The consensus core of component $j$ at stability threshold $\tau$ is $\text{core}_\tau(j) = \{g \in S_j^{(0)} : f_{jg} \geq \tau\}$ — the genes in the reference component that appear in at least $\tau$ fraction of the matched partners. The *stable support* at threshold $\tau$ is the union $\text{support}_\tau = \bigcup_j \text{core}_\tau(j)$.

*Step 5: Variance explained.* To assess how much signal the stable support carries, we zero out all rows of $\mathbf{W}$ not in $\text{support}_\tau$ and recompute the variance explained on the centered data $\tilde{\mathbf{X}}$, $\text{VE} = 1 - \|\tilde{\mathbf{X}} - \tilde{\mathbf{X}} \mathbf{W} \mathbf{W}^\top\|_F^2 / \|\tilde{\mathbf{X}}\|_F^2$, in minibatches over the sparse matrix. The pruned $\mathbf{W}$ enters as-is: we do not re-

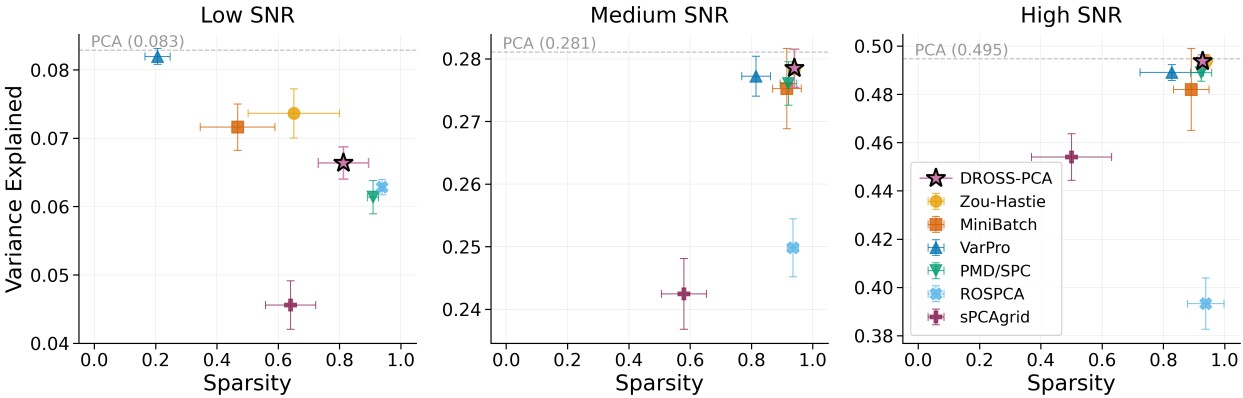

Figure 19: Sparsity vs. variance explained: S1c, 5% at 4×.

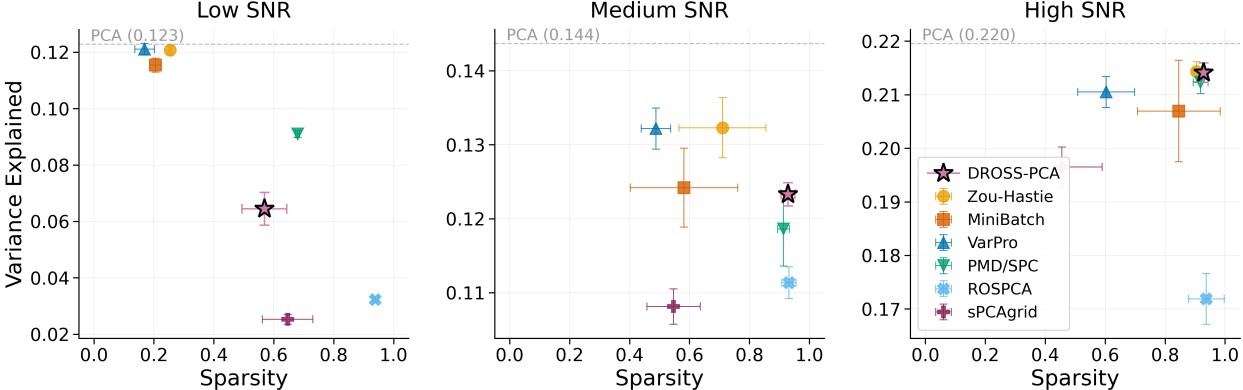

Figure 20: Sparsity vs. variance explained: S1c, 5% at 8×.

orthonormalize after pruning, so VE is a reconstruction fit, not a projection onto span(**W**). Because the learned loadings remain near-orthonormal after pruning (column norms within 0.4% of unity and columns pairwise within 0.2° of orthogonal), the tied-weight VE agrees with the orthogonal-projection VE to within 0.2 percentage points (81.05% vs. 81.20% at the full support).

**Functional enrichment.** Gene identifiers (Ensembl IDs) are mapped to gene symbols via BioMart. Over-representation analysis uses Fisher's exact test (GSEApy, Fang et al. 2022) against GO Biological Process 2023, KEGG 2021 Human, and Reactome 2022. Background is the set of 14,990 active genes with valid symbol mappings.

**Enrichment baselines.** To test whether the core enrichment is specific to the recovered modules rather than an artifact of gene-set size, we compared each of the 50 cores against three size-matched baselines under the identical over-representation test—same background (14,990 active genes), same GO/KEGG/Reactome libraries, same Benjamini–Hochberg cutoff: (i) random gene sets drawn from the active background (200 draws per core), (ii) the top-$s$ genes by absolute loading on the leading PCA components, and (iii) the top-$s$ highly variable genes (`scanpy`, Seurat flavor, Wolf et al. 2018), with $s$ matched to each core's size (Table 10). Random sets yielded a median of 0 significant terms (mean < 0.1), and every core exceeded the 95th percentile of its own random null ($p$-value < 0.005), confirming the signal is not a size effect. PCA top-loadings and highly variable genes both enriched strongly, as expected for structured selections and at comparable top-term strength ($\sim 10^{-29}$). The cores matched or exceeded PCA top-loadings for 52% of components and highly variable genes for 78%. The consensus cores are therefore as functionally coherent as dense PCA top-loadings and more coherent than a standard variability filter, while being the

**PCA Baseline Across Configurations and Noise Levels**

Figure 21: PCA baseline variance explained and reconstruction loss across all structural configurations and noise levels.

only selection that is simultaneously sparse, resolved at the component level, and defined by reproducibility across initializations.

Table 10: HLCA enrichment versus size-matched baselines. For the 50 largest consensus cores, the number of significantly enriched terms (Fisher's exact test, adjusted $p$-value $< 0.05$, against GO Biological Process, KEGG, and Reactome, with background $= 14{,}990$ active genes). Each baseline is matched to the corresponding core's size.

| Gene-set selection | Median sig. terms | Mean sig. terms | Top term ($-\log_{10}$ adj $p$) |
|---|---|---|---|
| DROSS-PCA consensus cores | 487 | 505 | 29.3 |
| PCA top-loadings | 522 | 541 | 29.2 |
| Highly variable genes | 386 | 427 | 29.7 |
| Random (size-matched) | 0 | $< 0.1$ | 0.1 |

Table 11: Variance explained by stable gene supports at different stability thresholds. Support-$\tau$: union of all genes appearing in $\geq \tau\%$ of matched components across 99 other runs. Full model: all 15,296 active genes.

| Threshold ($\tau$) | Support size | VE (%) | % of full VE |
|---|---|---|---|
| 0% (full) | 15,296 | 81.05 | 100.0 |
| 50% | 11,670 | 81.01 | 100.0 |
| 70% | 9,898 | 80.87 | 99.8 |
| 80% | 8,716 | 80.65 | 99.5 |
| 90% | 7,242 | 80.01 | 98.7 |
| 95% | 2,858 | 46.91 | 57.9 |
| 100% | 662 | 9.86 | 12.2 |

# E    Application Details: Sea-Surface Temperature Fields

**Data.**    Daily 00 UTC sea-surface temperature is taken from the ERA5 reanalysis (Hersbach et al., 2020) as distributed by WeatherBench-2 (Rasp et al., 2024), over 1959-01-01 to 2022-12-31 ($n = 23{,}376$ days) at the $1440 \times 721$ ($0.25°$, longitude $\times$ latitude) equiangular resolution, together with the matching static land–sea mask, the daily sea-ice-cover field, and the shipped 1990–2019 daily SST climatology.

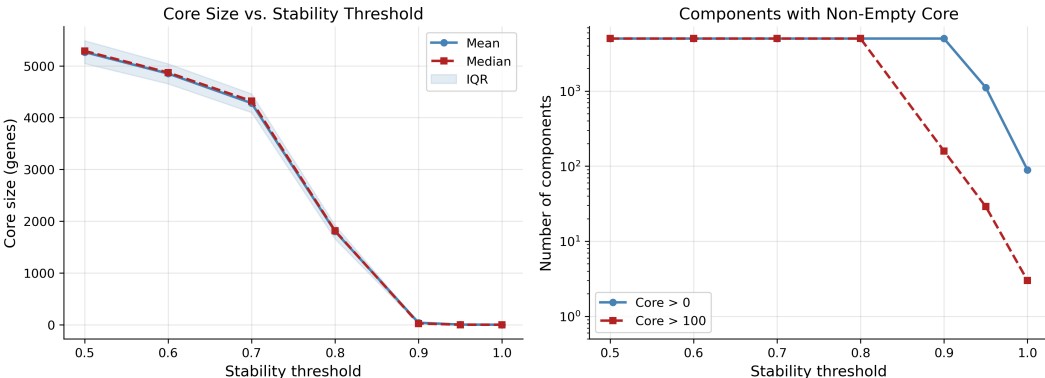

Figure 22: Distribution of consensus core sizes at the 90% stability threshold across 5,000 components. Left: all components (log $y$-axis). Right: nonzero cores only (log-log). Median core: 24 genes. In all, 159 components have cores exceeding 100 genes. Largest core: 1,883 genes.

Figure 23: Left: mean and median consensus core size as a function of stability threshold. Core sizes decline gradually from 80% to 90% and drop sharply above 90%. Right: number of components with non-empty cores (blue) and cores exceeding 100 genes (red).

**Preprocessing.** (i) *Deseasonalize*: subtract the shipped daily climatology indexed by day-of-year (leap days handled directly, day-of-year 1–366). (ii) *Mask* $\Omega$: keep cells that are ocean (fractional land–sea mask $< 0.5$) and never ice-covered (sea-ice fraction $\leq 0.15$ at every time step), leaving $p = 484{,}778$ ocean cells (46.7% of the $1440 \times 721$ grid). (iii) *Detrend*: remove a per-cell least-squares linear fit in time, keeping the residual, which also zero-means each column. (iv) *Area-weight*: scale each column by $\sqrt{\cos\phi}$ ($\phi$ the cell latitude), so the Frobenius geometry of $\mathbf{X}$ matches an area-weighted variance. Loadings are divided by $\sqrt{\cos\phi}$ before plotting. No per-feature standardization is applied—consistent with the covariance-form input used throughout, this lets high-variance regions (the equatorial and frontal Pacific) express in the leading modes rather than being rescaled to unit variance. The result is $\mathbf{X} \in \mathbb{R}^{n \times p}$ in single precision. At $0.25°$ it occupies $\sim45$ GB and is held once in host memory.

**Niño-3.4 reference index.** The observed Niño-3.4 index is the $\cos\phi$-weighted mean SST anomaly over the box latitude $[-5°, 5°]$, longitude $[190°, 240°]$ ($170°$W–$120°$W), from the deseasonalized field before detrending. It is used only for validation and never enters the fit. Component matching uses the maximum absolute Pearson correlation over the $k$ learned components.

**Dense-EOF ceiling and initialization.** Forming the $p \times p$ covariance is infeasible ($\sim1.9$ TB at $p = 484{,}778$), so the leading EOF subspace is computed by a matrix-free randomized SVD of $\mathbf{X}$ (Halko et al., 2011) (single precision, $k = 50$, 7 power iterations, oversampling 20). We report the *realized* variance explained,

$\|\mathbf{X}V\|_F^2/\|\mathbf{X}\|_F^2$ with orthonormal $V$, rather than the singular-value estimate. At $0.25°$ the cumulative realized VE is $12.9\%$ at $k = 1$, $49.2\%$ at $k = 20$, and $66.1\%$ at $k = 50$. The same $V$ (sliced to the run's $k$) is passed to DROSS-PCA as both the initialization and the calibration components, so the estimator never densifies $\mathbf{X}$ to seed itself.

**Training configuration.** The configuration is $k = 20$, robust weighting enabled, single precision, element-wise sparsity, batch size 512, 3,000 steps with 1,500 warm-up steps, sparsity and orthogonality weights $\lambda_s = \lambda_{\mathrm{orth}} = 0.5$, and the EOF initialization above. The fit reads $\mathbf{X}$ from disk in single precision and moves one minibatch to the accelerator at a time, so device memory scales with the batch, not with $p$. The fit takes $\sim$1,200 s on one GPU. Cosine-preserving pruning then sweeps $\theta \in [0°, 10°]$ (100 points, batch size 2,048) to produce the sparsity–variance frontier and select the operating point by the $d\theta/ds$ criterion (Section 3.2). At $0.25°$ this selects $\theta = 3.3°$ ($27\%$ sparse, $47.7\%$ variance explained).

**Localization metric.** For a loading vector $v$ over ocean cells with per-cell areas $a$, the effective number of active cells is the inverse participation ratio $N_{\mathrm{eff}} = (\sum_i v_i^2)^2/\sum_i v_i^4$, and its physical footprint is $N_{\mathrm{eff}} \cdot \bar{a}$ with $\bar{a}$ the energy-weighted mean cell area $\sum_i v_i^2 a_i/\sum_i v_i^2$. A mode concentrated on few cells has a small footprint. A basin-spanning mode has a large one. Figure 9 reports this footprint for the sparse loadings and, for comparison, for the dense EOFs. The sparse mean is $2.5 \times 10^7 \, \mathrm{km}^2$ versus $7.1 \times 10^7 \, \mathrm{km}^2$ for the dense EOFs.

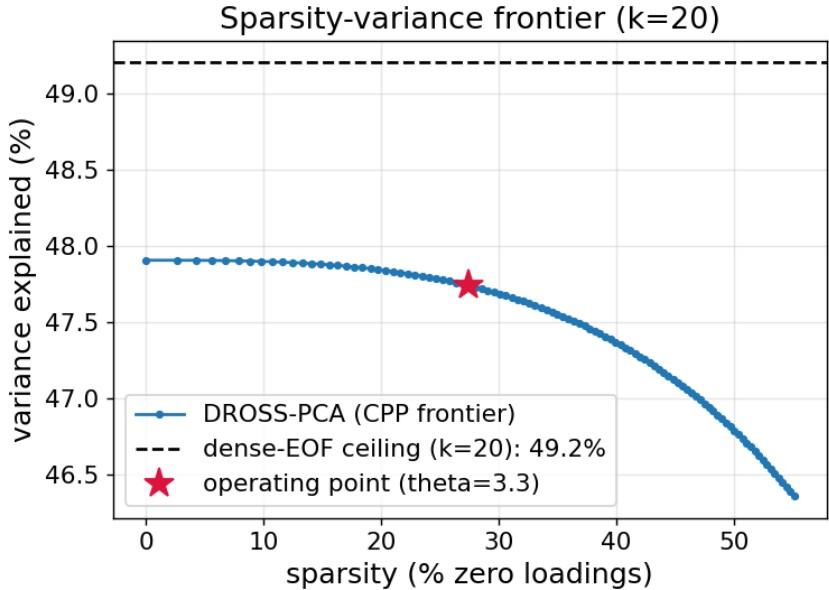

Figure 24: SST sparsity–variance frontier at $0.25°$. The CPP frontier of the learned $k = 20$ loadings (variance explained vs. fraction of zero loadings) against the dense-EOF ceiling (dashed). The star is the selected operating point ($\theta = 3.3°$, $27\%$ sparse, $47.7\%$ variance explained).

## F  Stability Analysis Details

The identifiability analysis (Section 4.3) runs 100 random orthonormal initializations per replicate across all four factorial configurations at three SNR levels (120,000 total fits). Global AUPRC and Q2–Q4 metrics reported in the main text are aggregated over 100 replicates (mean $\pm$ std). The full SNR breakdown including global AUPRC is in Table 8.

# G   Extended Discussion

**Smooth penalties and correlated features.**   With a smooth penalty, no coefficient is set exactly to zero during optimization. All coefficients evolve simultaneously along continuous gradients, which may allow correlated features to settle at similar magnitudes rather than being arbitrarily partitioned. This hypothesis is consistent with the stability results (Section 4.3), but a direct test on block-correlated features is left for future work.

**Relationship to robust sparse PCA methods.**   Recent methods combining robustness with sparsity (Croux et al., 2013; Hubert et al., 2016; Pfeiffer et al., 2025; Puchhammer et al., 2026) differ in both corruption model and computational strategy. ROSPCA (Hubert et al., 2016), sPCAgrid (Croux et al., 2013), and the MRCD-based approach of Puchhammer et al. (2026) target casewise (whole-sample) outliers and operate on the full data or covariance matrix, restricting them to $p \lesssim 1{,}000$. SCRAMBLE (Pfeiffer et al., 2025) instead addresses cellwise (individual-entry) corruption and uses minibatch Riemannian SGD, so it is distinguished from DROSS-PCA by corruption model rather than by batch structure. DROSS-PCA integrates robustness directly into a formulation that scales more than an order of magnitude beyond existing robust sparse PCA methods, while targeting the casewise model relevant to whole-sample biomedical artifacts.

**Relationship to the combinatorial literature.**   A distinct tradition formulates sparse PCA as a cardinality-constrained optimization on the $p \times p$ covariance matrix (d'Aspremont et al., 2007; Berk and Bertsimas, 2019; Bertsimas et al., 2022; Del Pia et al., 2025). These methods require $O(p^2)$ storage and address a single sparse direction at a time. The Fantope relaxation (Vu et al., 2013) and its stochastic variant (Qiu et al., 2023) are closest: both use minibatch gradient descent for genomics, but optimize a $p \times p$ PSD matrix ($O(p^3)$ per step), while DROSS-PCA optimizes the $p \times k$ loading matrix ($O(bpk)$ per step). At the $p = 27{,}402$ of our single-cell data, the $O(p^3)$ projection becomes prohibitive.

