# OpenReview forum: "Scale-Invariant, Robust Sparse PCA for Large Data via Differentiable Penalties"
_TMLR — Under review for TMLR_

### Review · Reviewer_AhXj · 2026-04-25

**Summary Of Contributions:**

The authors propose an innovative approach to sparse dimension reduction that decouples learning of the components and feature selection, avoiding the early binary elimination of some features, and uses a differentiable objective, allowing acceleration with GPUs, and scalability to larger datasets. They illustrate interesting properties of their methods, in terms of stability of variable selection and obtained components, and robustness to outliers.

**Audience:**

Yes

**Audience Explanation:**

The method offers a clear opportunity for a broader applicability of sparse PCA methods, while maintaining good performances compared to competitors. However, it would need a github repository, with a tutorial and clear installation instructions to be actually usable. The code in supplementary is not really documented enough.

**Broader Impact Concerns:**

no concern

**Claims And Evidence:**

Yes

**Claims Explanation:**

The rationale of ROSS-PCA is well explained, and a variety of experiments illustrate practical aspects of the method, and the real data application provides good insights of the method's applicability in practice. One claim that is not well illustrated is that the CPU fallback of ROSS-PCA is also competitive, it would be interesting to add it in Figure 5.

**Requested Changes:**

As mentioned in my comments, the main changes would be
- add the computation time for ROSS-PCA on CPUs. It is really important as not everyone has GPUs. Also what is the impact on the dataset size that can be computed under 6 hours?
- improve the code sharing and documentation otherwise it will not be really usable by another person, especially in the applicative fields, such as biology etc
- metrics should all be described briefly in the manuscript

---

> ### Author Response · Authors · 2026-07-16
>
> **AhXj.1 — Add CPU compute time for DROSS-PCA (+ data set size feasible under 6 h).**
> The scaling figure now reports CPU wall-clock alongside the GPU curve: on CPU, DROSS-PCA
> completes the largest benchmark ($p = 10{,}000$, $n = 100{,}000$, $k = 200$) in ${\sim}53$ min
> — within a 6-hour budget — and grows more slowly with $p$ than MiniBatch SparsePCA. Details
> are in the *Computational scaling* section (Section 4.4).
>
> **AhXj.2 — Improve code sharing and documentation (repo, tutorial, install).**
> We include the full DROSS-PCA implementation as **anonymized supplementary material**: a
> documented Python package (the exact method + cosine-preserving-pruning code), a README
> with installation and API, and a runnable **tutorial notebook** (`examples/tutorial.ipynb`)
> covering fitting, CPP pruning, support recovery, robustness, and the stability diagnostic. We
> keep it anonymized rather than a public repository to preserve double-blind review. A public
> release will follow after journal's final decision.
>
> **AhXj.3 — Describe all metrics briefly in the manuscript.**
> We audited every plotted metric against its first use. Most were already defined at first
> mention. We added brief main-text glosses for the three previously defined only in the
> appendix — *sparsity*, *relative reconstruction error*, and *Jaccard similarity* — in the
> *Evaluation Metrics* section (Section 3.4), which points to Appendix B.2 for full formulas.

---

### Review · Reviewer_5j3w · 2026-06-18

**Summary Of Contributions:**

The paper proposes ROSS-PCA, a novel method for sparse PCA. The high level approach is at follows: the authors first optimize a smooth differentiable objective with three terms (reconstruction error, sparsity, and orthogonality of factors). Then, there is a post-hoc cosine-preserving pruning step to obtain sparse loadings. The main benefits are scalability, robustness to outliers, and stability analysis via multiple random initializations. The authors show results on synthetic sparse PCA benchmarks, outlier settings, and a large-scale experiment on the Human Lung Cell Atlas.

**Audience:**

No

**Audience Explanation:**

Sparse PCA is a very mature field, with hundreds of papers (and the authors cite many of them). The methodological contribution here  relative to this body of work appears to be incremental. The algorithmic ingredients of ROSS-PCA are all standard -- smooth nonconvex optimization, sparsity-promoting penalties, pruning, random restarts, stability analysis --- all of them have been explored in various incarnations before. I do not see a fundamentally new ML idea or finding that will be broadly interesting.

The strongest contribution is practical scalability for sparse matrices arising from biology. That may interest a subset of readers working on single-cell genomics. But as written, the paper does not establish broad relevance. On clean synthetic benchmarks, the method is mostly competitive with (but not clearly better than) classical sparse PCA methods.

Overall, I find the paper technically solid but not sufficiently interesting to the broader TMLR audience.

**Broader Impact Concerns:**

None to report.

**Claims And Evidence:**

Yes

**Claims Explanation:**

The proposed method is reasonably intuitive but not supported with theoretical analysis --- so the main evidence is empirical/experimental and I will evaluate it as such.

On the experiments side, the authors do a reasonably thorough job. The writing is clear and everything seems technically correct (barring some minor comments, see requested changes below). Moreover, ROSS-PCA is shown to be competitive with established methods on synthetic benchmarks, while being more robust to outliers. One thing that the authors could perhaps do is to include a couple more large-scale experiments (not just the one on the Human Lung Cell Atlas).

**Requested Changes:**

* Some claims need to be toned down. Eg p1 "**Every** sparse PCA does both simultaneously" seems either untrue or hard to justify rigorously.
* Consider explicitly positioning the paper as a scalable method for very large, sparse biological matrices.
* Cosine-preserving pruning and knee selection are the main heuristics. Consider showing sensitivity to the knee rule, smoothing, grid resolution, number of random starts, and threshold.
* Add 1 or 2 more large scale experiments.

---

> ### Author Response · Authors · 2026-06-23
> **Request for a concrete baseline**
>
> Thank you for the thoughtful review. We will wait for all reviews before posting a revision, but wanted to raise one question now.
>
> Your comment notes that the ingredients of ROSS-PCA have appeared "in various incarnations," and we would like to benchmark against the closest prior method. The defining choice in ROSS is to **defer** selection out of the optimizer — train a dense loading matrix to convergence and impose exact zeros only afterward — and in surveying the field we could not find a prior method that does this, nor the two capabilities this design enables (see after the table). So we would value a concrete pointer. The table is what our search turned up. (Staged procedures such as iterative thresholding decouple the two *tasks* but still threshold *within* the iteration, so they do not defer in this sense.)
>
> | Method / family (representative) | Deferred | Scales | Robust | Calib. | Init. stab. |
> |---|:--:|:--:|:--:|:--:|:--:|
> | SDP relaxation (d'Aspremont et al., 2007) | ✗ | ✗ | ✗ | ✗ | ✗ |
> | Certifiable MIP (Berk & Bertsimas 2019; Bertsimas et al. 2022; Del Pia et al. 2025) | ✗ | ✗ | ✗ | ✗ | ✗ |
> | SCoTLASS (Jolliffe et al., 2003) | ✗ | ✗ | ✗ | ✗ | ✗ |
> | SPCA / elastic net (Zou et al., 2006) | ✗ | ✗ | ✗ | ✗ | ✗ |
> | Regularized SVD (Shen & Huang, 2008) | ✗ | ✗ | ✗ | ✗ | ✗ |
> | PMD / SPC (Witten et al., 2009) | ✗ | ✗ | ✗ | ✗ | ✗ |
> | MiniBatch SparsePCA (Pedregosa et al., 2011) | ✗ | ◐ | ✗ | ✗ | ✗ |
> | Variable projection (Erichson et al., 2020) | ✗ | ◐ | ◐ † | ✗ | ✗ |
> | Iterative thresholding (Ma 2013; Johnstone & Lu 2009) | ✗ | ✗ | ✗ | ✗ | ✗ |
> | Fantope FPS (Vu et al., 2013) | ✗ | ✗ | ✗ | ✗ | ✗ |
> | Gradient / online (Qiu et al., 2023) | ✗ | ◐ ‡ | ✗ | ✗ | ✗ |
> | Robust sparse PCA (Croux et al. 2013; Hubert et al. 2016; Pfeiffer et al. 2025) | ✗ | ✗ | ✓ | ✗ | ✗ |
> | Stability selection (Meinshausen & Bühlmann 2010; Sill et al. 2015) | ✗ | ✗ | ✗ | ✗ | ◐ § |
> | **ROSS-PCA (ours)** | **✓** | **✓** | **✓** | **✓** | **✓** |
>
> **Marks:** ✓ yes · ◐ partial · ✗ no.
> **Columns:** *Deferred* = trains a dense loading matrix and zeros entries only afterward; *Scales* = GPU/minibatch usable at p ≳ 10⁴ on sparse matrices; *Robust* = built-in outlier resistance; *Calib.* = penalty weights normalized to a transferable, dimensionless scale; *Init. stab.* = identifiability diagnostic from random *initializations* at the *component* level.
> † robust variant exists but is not the default; ‡ minibatch/online, but optimizes a p×p projection (O(p³)/step), prohibitive at p ≈ 27,000; § stability diagnostic, but by resampling the *data* at the *feature* level, not from *initializations* at the *component* level.
>
> No single ingredient here is new, and our point is not that. It is that we could find no method occupying the same point. Two capabilities in particular have no precedent we could find — the initialization-based stability diagnostic (**Init. stab.**) and the reconstruction-floor calibration (**Calib.**) — for two *different* reasons:
>
> - **Init. stab.** seems unavailable to the dominant paradigm by construction: binary keep-or-zero methods eliminate features irrecoverably, so restarts cannot re-explore support, and convex relaxations (Fantope; Qiu et al., 2023) have a unique optimum, so initialization carries no information. The closest analogue, stability selection / S4VDPCA, resamples the *data* and scores *feature* inclusion — a different perturbation and a different unit. ROSS makes the *initialization itself* informative, at the component level.
>
> - Our specific **Calib.** has, to our knowledge, no precedent in sparse PCA. By the Eckart–Young–Mirsky theorem the rank-k PCA reconstruction loss is the minimum reconstruction error over all rank-k projections, so normalizing the additive penalties by it sets penalty pressure relative to a *provable* floor, making the weights dimensionless and transferable. glmnet's λ_max is similar in spirit but anchors at the *null* solution and is sparse *regression*, not PCA.
>
> If there is a specific algorithm you have in mind — especially one that defers selection, or performs initialization-based component-level stability — we will add it as a baseline and benchmark where feasible. We are also happy to extend the comparison along any axis you think is missing.

---

> > ### Comment · Reviewer_5j3w · 2026-07-02
> > **Thanks**
> >
> > A table like this would be very helpful (and immediately highlights the key contributions) -- in lieu of the rather short paragraph on comparisons to prior work -- please consider including it in your revision. Looking forward to the next version of the manuscript.

---

> ### Author Response · Authors · 2026-07-16
>
> **5j3w.1 — Tone down claims (e.g., p1 "Every sparse PCA does both simultaneously").**
> Softened to *"The sparse PCA methods we survey couple them…"* (Introduction, Section 1).
> The new Related Work section (Section 2) and its capability table give a per-method accounting.
>
> **5j3w (follow-up) — Include the comparison table in the revision.**
> Done — the full capability comparison table is now the core of the new **Related Work** section
> (Section 2), and the old "closest prior work" paragraph is trimmed to a pointer. This makes the
> axes of contribution explicit, including the two we identify as new (calibration and the
> initialization-based stability diagnostic).
>
> **5j3w.2 — Explicitly position as a scalable method for very large, sparse biological matrices.**
> We make the large-data positioning explicit in the Introduction (Section 1) and Related Work
> (Section 2), with one clarification: **DROSS-PCA is not restricted to sparse input matrices.**
> Its scalability comes from minibatching (it densifies one minibatch at a time), not from the
> input being sparse — input sparsity only reduces the resident memory. The new WeatherBench-2
> SST application (5j3w.4) fits a **fully dense** matrix ($p = 484{,}778$, ${\sim}45$ GB held once
> in host memory), so the method extends beyond sparse biological data to dense scientific
> fields, which also speaks to the broad-relevance concern in the review. We added one
> sentence making this explicit in the SST section (Section 4.6).
>
> **5j3w.3 — Sensitivity to knee rule, smoothing, grid resolution, #random starts, threshold.**
> The five knobs fall into three groups. The **knee rule, curve smoothing, and threshold** are
> one choice — where to stop on the CPP frontier — which we now scope as a limitation (Section 5,
> Discussion, *Limitations and future work*): the frontier itself is well-defined (CPP bounds
> each loading's angular deviation from the original, up to $\theta_{\max}=10^\circ$), but no
> universal criterion selects a *point* on it, and the elbow's sensitivity to frontier shape
> and to the Savitzky–Golay smoothing inside the knee detector is demonstrated by the S1c
> operating-point analysis (Section 4.1). **Number of random starts** is exactly what the
> stability analysis measures (Section 4.3, 100 initializations per configuration). **Grid
> resolution** only sets how finely the continuous frontier is sampled ($100$ points over
> $0$–$\theta_{\max}$ by default). It feeds the same knee detector as the smoothing and
> threshold, so its effect falls under the operating-point-selection sensitivity above.
>
> **5j3w.4 — Add 1–2 more large-scale experiments.**
> We added a second large real-data application in a different modality: daily ERA5 sea-surface
> temperature via WeatherBench-2 (new *Application: Sea-Surface Temperature Fields* section,
> Section 4.6, plus Appendix E), the full 1959–2022 daily record at $0.25^\circ$ ($p = 484{,}778$,
> comparable in scale to HLCA). The section reports three checks against an accepted physical
> reference: ENSO recovery ($|r| = 0.977$ vs. $0.923$ for the leading dense EOF), basin de-mixing
> (${\sim}2.8\times$ more localized than dense EOFs), and near-free sparsity (${\sim}27\%$ sparse
> at $47.7\%$ VE against a $49.2\%$ dense ceiling). DROSS-PCA is now validated on two large,
> structurally different data sets — one sparse, one dense.

---

### Review · Reviewer_PQX2 · 2026-07-03

**Summary Of Contributions:**

This paper proposes ROSS-PCA, a robust and scalable sparse PCA method. The key idea is to decouple learning and feature selection: the method first optimizes a smooth differentiable objective with reconstruction, sparsity, orthogonality, and robust weighting terms, then applies cosine-preserving pruning to obtain sparse loadings. The paper evaluates the method on synthetic sparse PCA benchmarks, outlier-contaminated data, scaling experiments, and the Human Lung Cell Atlas.

**Audience:**

Yes

**Audience Explanation:**

A GPU-scalable sparse PCA method with stability analysis is relevant to readers interested in interpretable dimensionality reduction, robust ML, and large-scale scientific data analysis.

**Claims And Evidence:**

No

**Claims Explanation:**

The paper is clearly written and the experiments are extensive. The scalability result is compelling, and the synthetic results show that ROSS-PCA is competitive with existing sparse PCA methods, especially under strong outlier contamination.

However, some claims are stronger than the evidence supports. The robustness claim is mainly shown against non-robust baselines; robust sparse PCA baselines are missing. The scale-invariance claim also needs more justification, since the robust sample weights appear to depend on raw residual magnitude. The variance-explained computation after pruning should be clarified because pruned loadings may no longer be orthonormal.

**Requested Changes:**

The authors should add robust sparse PCA baselines on the outlier experiments. They should clarify how variance explained is computed after pruning, and whether W is re-normalized, re-orthogonalized, or treated as a non-orthogonal basis.

They should provide stronger evidence for scale-invariant hyperparameters, ideally by testing rescaled datasets.

The HLCA biological validation should include stronger baselines, such as PCA loadings, highly variable genes, NMF, or size-matched random gene sets.

The paper should soften broad claims such as “all existing sparse PCA methods” and clarify that the full pipeline is not fully differentiable because pruning is post-hoc and discrete.

---

> ### Author Response · Authors · 2026-07-16
>
> **PQX2.1 — Add robust sparse PCA baselines on the outlier experiments.**
> We added **ROSPCA** (Hubert et al., 2016) and **sPCAgrid** (Croux et al., 2013) — both
> matching the S1c casewise-contamination model — and ran them across the full synthetic
> suite. The results and an operating-point-selection analysis are in Section 4.1 (Results),
> under the paragraph *S1c: robustness to outlier contamination*. Consistent with them, the
> abstract and robustness discussion now state that DROSS-PCA substantially exceeds non-robust
> methods and remains competitive with the dedicated robust methods, which do not scale (see
> 5j3w.2), replacing the earlier "particular advantage under severe contamination".
>
> **PQX2.2 — Clarify how variance explained is computed after pruning.**
> Clarified in Appendix D (*Step 5: Variance explained*): VE is the tied-weight
> reconstruction $R^2$ with the pruned $W$ used **as-is** — not re-normalized or
> re-orthogonalized. We verify there that the pruned loadings remain near-orthonormal, so the
> tied-weight and orthogonal-projection VE coincide, and the reported VE is not an artifact of
> the tied-weight choice.
>
> **PQX2.3 — Stronger evidence for scale-invariant hyperparameters (test rescaled data).**
> We added a dedicated *Invariance to data scale* subsection (Section 4.2) giving both the
> construction-level argument (every objective term is scale-free) and a direct rescaling
> experiment over six orders of magnitude on clean and contaminated data. On clean data the fit
> is invariant to machine precision, operating point included. On contaminated data the
> scale-free argument still governs the **continuous minimizer** — recovery, VE, and loadings
> stay invariant to within $4\times10^{-3}$ (loadings $<0.1^\circ$) — but the **separate discrete
> pruning step** selects its operating point by a knee rule that is ill-conditioned on the flat
> contaminated frontier, so the reported sparsity and threshold can shift across scales at
> essentially the same recovery. That is the operating-point-selection sensitivity we scope as a
> limitation (5j3w.3). Numbers are in that subsection.
>
> **PQX2.4 — Stronger HLCA biological baselines.**
> We added the requested size-matched baselines — random gene sets, PCA top-loadings, and
> highly variable genes — under an identical over-representation test, with a new supplementary
> table and supporting text in the single-cell section (Section 4.5) and Appendix D. We
> did not include NMF: it is a separate full factorization rather than a size-matched selection,
> and prohibitive at this scale — the three included baselines already isolate the
> gene-set-size concern raised.
>
> **PQX2.5a — Soften broad claims ("all existing sparse PCA methods").**
> Softened in the Introduction (Section 1, now *"The sparse PCA methods we survey couple them…"*,
> scoped and pointing to the comparison table), and the blanket prose is replaced by the
> per-method capability table in the new Related Work section (Section 2).
>
> **PQX2.5b — Clarify that the full pipeline is not fully differentiable (pruning is discrete).**
> Made explicit right after the two-stage-pipeline description (Section 3, Methods): the
> pruning stage is discrete, so the pipeline is not end-to-end differentiable — the title's
> "differentiable" refers to the learning *penalties*, and keeping selection out of the
> differentiable objective is what enables the stability analysis.

---

### Author Response · Authors · 2026-07-16
**Responses to reviewers and revised manuscript submitted**

We thank the reviewers for their constructive feedback. We have replied to each reviewer, pointing to the revised manuscript for details. We look forward to the discussion and to addressing any remaining concerns.